Predicted Ultrafine Particulate Matter Source Contribution across the Continental United States during Summer Time Air Pollution Events

Melissa A. Venecek[a], Xin Yu[b] and Michael J. Kleeman[b]

[a]*Department of Land, Air and Water Resources, University of California Davis, Davis, CA*
[b]*Department of Civil and Environmental Engineering, University of California Davis, Davis, CA*

*Corresponding author. Tel.: +1 530 752 8386; fax; +1 530 752 7872. E-mail address: mjkleeman@ucdavis.edu (M.J. Kleeman).

**Abstract**

The regional concentrations of airborne ultrafine particulate matter mass ($Dp < 0.1$ µm; $PM_{0.1}$) were predicted in 39 cities across the United States (U.S.) during summer time air pollution episodes. Calculations were performed using a regional source-oriented chemical transport model with 4 kilometer (km) spatial resolution operating on the National Emissions Inventory created by the U.S. EPA. Measured source profiles for particle size and composition between $0.01 – 10$ µm were used to translate PM total mass to $PM_{0.1}$. Predicted $PM_{0.1}$ concentrations exceeded 2 µg/m$^3$ during summer pollution episodes in major urban regions across the U.S. including Los Angeles, the San Francisco Bay Area, Houston, Miami, and New York. $PM_{0.1}$ spatial gradients were sharper than $PM_{2.5}$ spatial gradients due to the dominance of primary aerosol in $PM_{0.1}$. Artificial source tags were used to track contributions to primary $PM_{0.1}$ and $PM_{2.5}$ from fifteen source categories. On-road gasoline and diesel vehicles made significant contributions to regional $PM_{0.1}$ in all 39 cities even though peak contributions within 0.3 km of the roadway were not resolved by the 4 km grid cells. Food cooking also made significant contributions to $PM_{0.1}$ in all cities but biomass combustion was only important in locations impacted by summer wildfires. Aviation was a significant source of $PM_{0.1}$ in cities that had airports within their urban footprints. Industrial sources including cement manufacturing, process heating, steel foundries, and paper & pulp processing impacted their immediate vicinity but did not significantly contribute to $PM_{0.1}$ concentrations in any of the target 39 cities. Natural gas combustion made significant contributions to $PM_{0.1}$ concentrations due to the widespread use of this fuel for electricity generation, industrial applications, residential and commercial use. The major sources of primary $PM_{0.1}$ and $PM_{2.5}$ were notably different in many cities. Future epidemiological studies may be able to differentiate $PM_{0.1}$ and $PM_{2.5}$ health effects by contrasting cities with different ratios of $PM_{0.1}$ / $PM_{2.5}$. In the current study, cities with higher $PM_{0.1}$ / $PM_{2.5}$ ratios (ratio greater than 0.10) include Houston TX, Los Angeles CA, Bakersfield CA, Salt Lake City UT, and Cleveland OH. Cities with lower $PM_{0.1}$ to $PM_{2.5}$ ratios (ratio lower than 0.05) include Lake Charles LA, Baton Rouge LA, St. Louis MO, Baltimore MD, and Washington DC.

## 1. Introduction

Airborne particulate matter (PM) has been linked with premature mortality and numerous other health risks in cities across the world (see for example references (Dominici et al., 2006;Franklin et al., 2007;Pope et al., 2002;Ostro et al., 2006;Pope et al., 2009;Laden et al., 2000;Kheirbek et al., 2013;Aneja et al., 2017)). Despite years of progress (EPA, 2017b), PM concentrations in many urban regions in the U.S. still exceed health-based standards resulting in an increase of non-accidental mortality (Franklin et al., 2007;Baxter et al., 2013). Toxicology testing suggests that ultrafine particles with diameter < 0.1 μm may be the most harmful size fraction within $PM_{2.5}$ (Li et al., 2003;Oberdurster, 2000;Ostro et al., 2015;Oberdorseter et al., 1995;Pekkanen et al., 1997). Initial attempts to analyze ultrafine particles in epidemiology studies have used particle number concentration as a surrogate for ultrafine particle exposure, but this approach has not found consistent relationships with health effects (HEI, 2013). In contrast, a recent epidemiology study based on ultrafine particle mass ($PM_{0.1}$) found significant associations with premature mortality (Ostro et al., 2015). In addition, ultrafine (UF) mass concentrations are highly correlated with particle surface area and can be a good metric for the potential exposure to UF particles (Kuwayama et al., 2013;Ostro et al., 2015) . Follow-up studies have also found significant associations between $PM_{0.1}$ and reproductive outcomes including low birth weight and preterm birth (Laurent et al., 2016;Bergin et al., 1996). These findings have biological plausibility, since ultrafine particles may cross cell membranes and interfere with internal cell function (Sioutas et al., 2005). Ultrafine particles have greater surface area-per volume due to the small particle diameter making it more available for chemical reaction. Ultrafine particles can therefore have a larger impact when deposited deep into the lung cavity where they are not easily removed (Nel et al., 2006;Li et al., 2003).

A monitoring network for $PM_{10}$ and $PM_{2.5}$ has been operating throughout the continental U.S. for almost 20 years. Multiple studies have performed source apportionment calculations for coarse and fine PM using these measurements (Reff et al., 2009;Zhang et al., 2014;Zheng et al., 2002;Ham and Kleeman, 2011). In contrast, measurements of $PM_{0.1}$ are limited to focused field campaigns lasting for short time periods with even fewer studies attempting source apportionment calculations (Kleeman et al., 2009). Multiple barriers have prevented the widespread deployment of $PM_{0.1}$ monitoring network including (i) the low concentration of

$PM_{0.1}$ mass, which challenges the detection limits of analytical methods, (ii) the artifacts associated with collecting $PM_{0.1}$ samples, (iii) the additional workload involved in operating the collection devices, and (iv) the sharp spatial gradients of $PM_{0.1}$ concentrations. Expensive investments in $PM_{0.1}$ monitoring are unlikely to occur without compelling evidence linking $PM_{0.1}$ to public health. Early epidemiological studies for $PM_{0.1}$ must therefore use some other technique besides direct measurements to calculate population exposure.

Various methods such as the source-resolved PMCAMx chemical transport model, the chemical mass balance (CMB) model, photochemical box models and land use regression (LUR) models have been used to track source contributions to primary organic matter, elemental carbon and in some cases particle number concentration ($N_x$) over areas in the Eastern U.S. and parts of Europe and Asia (Lane et al., 2007;Posner and Pandis, 2015;Wang et al., 2011;Cattani et al., 2017;Wolf et al., 2017;Simon et al., 2018;Gaydos et al., 2005;Zhong et al., 2018). However, these methods are limited in one or more aspects of their ability to predict population exposure to ultrafine particles over large analysis domains. Source resolved models, such as PMCAMx, have been used to resolve composition for $N_x$ in the Eastern U.S. but not for $PM_{0.1}$ (Posner and Pandis, 2015). CMB models need measurements of specific molecular markers at numerous sites to resolve the sharp spatial gradients of ultrafine particle source contributions. LUR models need comprehensive measurements that act as training data sets in order to extend throughout a modeling domain (Lane et al., 2007).

Hu et al. (Hu et al., 2014) calculated population exposure to $PM_{0.1}$ in California using a regional source-oriented chemical transport model supported by measured profiles for particle size and composition of particles emitted by dominant sources. Predictions were compared to all available fine and ultrafine particle measurements over the period 2000-2010 with good agreement observed for the dominant chemical components of $PM_{0.1}$ mass including organic aerosol, elemental carbon, and numerous trace metals (Hu et al., 2014). The 4km spatial resolution used in these calculations supported multiple epidemiological studies based on spatial gradients of exposure (Ostro et al., 2015;Laurent et al., 2016). These encouraging results motivate the expansion of the $PM_{0.1}$ exposure technique to other locations.

Here we use the Eulerian source-oriented UCD/CIT chemical transport model to predict the concentration of $PM_{0.1}$ in thirty-nine urban regions throughout the U.S. during summer

pollution events in 2010. The calculation tracks contributions from fifteen (15) primary particle sources through a simulation of all major atmospheric processes while retaining information about particle size, composition and source origin (Hu et al., 2014). The results of this calculation reveal U.S. national trends in $PM_{0.1}$ concentrations for the first time and suggest locations where the differential health effects of $PM_{0.1}$ and $PM_{2.5}$ can best be studied.

## 2. Methods

### 2.1 Simulation Dates

Thirty-nine of the largest cities in the continental U.S. were selected as the primary target locations in the current study (Fig 1). These cities have been used to characterize atmospheric reactivity across the U.S. in previous air pollution studies (Carter, 1994a;Carter, 2010;Venecek et al., 2018a;Venecek et al., 2018b). Simulations within each target city were carried out during peak summer air pollution events in 2010. Dates were selected based on an initial investigation of measured 1-hr ozone ($O_3$) across all monitors in a core-based statistical area (CBSA). A CBSA is defined as a U.S. geographical area that consists of one or more counties anchored by an urban center of at least 10,000 people plus adjacent counties that have a high degree of social and economic integration with the core as measured by commuting (United States Census Bureau, 2018).

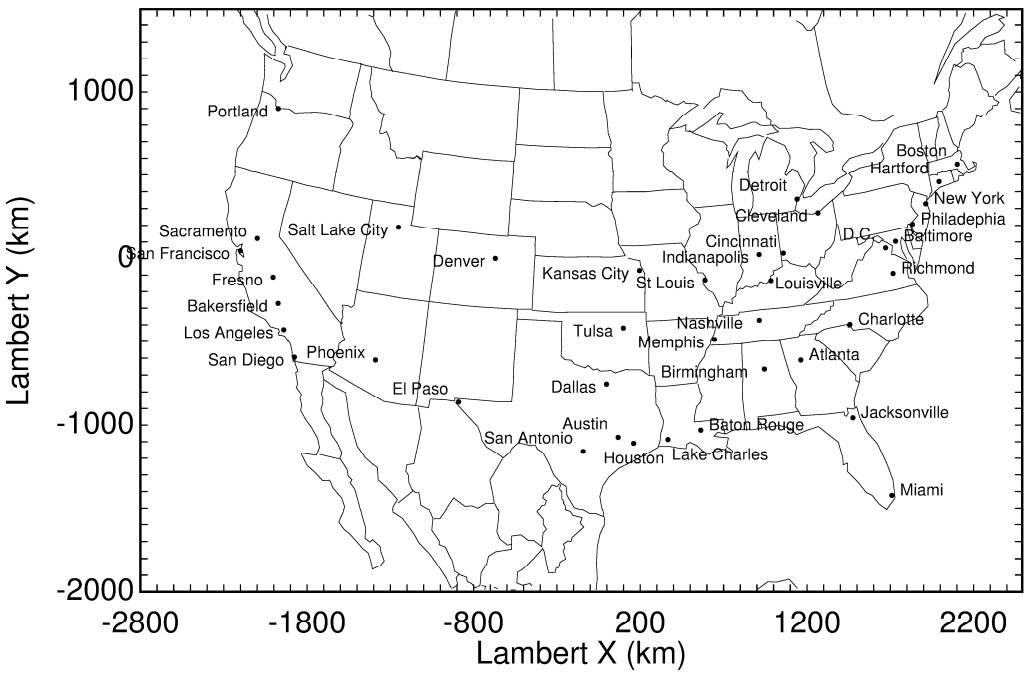

Figure 1 Map of 39 cities used for prediction of PM$_{2.5}$ and PM$_{0.1}$ source contribution across the continental United States during summer time air pollution events

The selected air pollution events within each CBSA typically had measured 1-hr maximum O$_3$ concentrations greater than 70 ppb. Regional pollution events caused by atmospheric stagnation were selected whenever possible as opposed to special events caused by unusual occurrences such as wildfires that affected only one city at a time. The simulation dates in each city are listed in Table 1.  Figure 2 illustrates the average 1-hr maximum O$_3$ concentration across all monitors within each CBSA during the selected regional events. Simulation periods are organized in chronological order for the year 2010 and cities within the same geographical region are grouped together.  Measured 24-hr PM$_{2.5}$ concentrations during peak summer pollution events ranged between 3.2-30 µg/m$^3$ depending on the location. The aggregation of these events across the U.S. enables a comparison of typical summer time air pollution episodes within different cities.

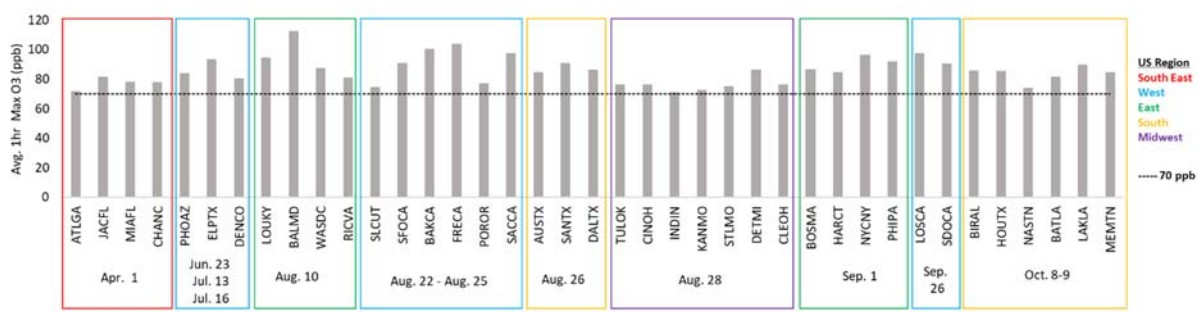

Figure 2 Average 1-hr Maximum O₃ across all monitors in each domain. Cities are grouped by corresponding extreme O₃ date (that was avg. > 70 ppb) and U.S. geographical region.

Table 1 City, City Code, Simulation Date, 2010 Population and Geographical Region

| City | City Code | 2010 Simulation Date[+] | 2010 Population | US Geographical Region |
|---|---|---|---|---|
| Atlanta | ATLGA | March 29 - April 1 | 422765 | South East |
| Austin | AUSTX | August 23 - August 26 | 815260 | South |
| Bakersfield | BAKCA | August 23 - August 26 | 348938 | West |
| Baltimore | BALMD | August 7 - August 10 | 621210 | East Coast |
| Baton Rouge | BATLA | October 6 - October 9 | 229584 | South |
| Birmingham | BIRAL | October 6 - October 9 | 212107 | South East |
| Boston | BOSMA | August 29 - September 1 | 620451 | East Coast |
| Charlotte | CHANC | March 30 - April 2 | 738710 | South East |
| Cincinnati | CINOH | August 25 - August 28 | 296904 | Midwest |
| Cleveland | CLEOH | August 25 - August 28 | 396009 | Midwest |
| Dallas | DALTX | August 23 - August 26 | 1201000 | South |
| Denver | DENCO | July 13 - July 16 | 603421 | West |
| Detroit | DETMI | August 25 - August 28 | 711299 | Midwest |
| El Paso | ELPTX | July 11 - July 14 | 651665 | West |
| Fresno | FRECA | August 23 - August 26 | 497090 | West |
| Hartford | HARCT | August 29 - September 1 | 125312 | East Coast |
| Houston | HOUTX | October 6 - October 9 | 2103000 | South |
| Indianapolis | INDIN | August 25 - August 28 | 830952 | Midwest |
| Jacksonville | JACFL | March 29 - April 1 | 823291 | South East |
| Kansas City | KANMO | August 25 - August 28 | 460639 | Midwest |
| Lake Charles | LAKLA | October 6 - October 9 | 72268 | South |
| Los Angeles | LOSCA | September 23 - September 26 | 3796000 | West |

| City | City Code | 2010 Simulation Date[+] | 2010 Population | US Geographical Region |
|---|---|---|---|---|
| Louisville | LOUKY | August 7 - August 10 | 300000 | Midwest |
| Memphis | MEMTN | October 6 - October 9 | 647609 | Midwest |
| Miami | MIAFL | March 30 - April 2 | 400769 | South East |
| Nashville | NASTN | October 7 - October 10 | 1800000 | Midwest |
| New York City | NYCNY | August 29 - September 1 | 8190000 | East Coast |
| Philadelphia | PHIPA | August 27 - August 30 | 1529000 | East Coast |
| Phoenix | PHOAZ | June 19 - Jun3 22 | 1449000 | West |
| Portland | POROR | August 23 - August 26 | 585286 | West |
| Richmond | RICVA | August 7 - August 10 | 204351 | East Coast |
| Sacramento | SACCA | August 22 - August 25 | 466488 | West |
| Salt Lake City | SLCUT | August 18 - August 21 | 186505 | West |
| San Antonio | SANTX | August 23 - August 26 | 1334000 | South |
| San Diego | SDOCA | September 23 - September 26 | 1306000 | West |
| San Francisco | SFOCA | August 22 - August 25 | 805704 | West |
| St. Louis | STLMO | August 25 - August 28 | 319257 | Midwest |
| Tulsa | TULOK | August 25 - August 28 | 392443 | Midwest |
| Washington DC | WASDC | August 7 - August 10 | 604453 | East Coast |

## 2.2 Model Description

The UCD/CIT model predicts the evolution of gas and particle phase pollutants in the atmosphere in the presence of emissions, transport, deposition, chemical reaction and phase change (Held et al., 2005) as represented by Eq. (1)

$$\frac{\partial C_i}{\partial t} + \nabla \cdot u C_i = \nabla K \nabla C_i + E_i - S_i + R_i^{gas}(C) + R_i^{part}(C) + R_i^{phase}(C) \qquad \text{(E1)}$$

where $C_i$ is the concentration of gas or particle phase species $i$ at a particular location as a function of time $t$, $u$ is the wind vector, $K$ is the turbulent eddy diffusivity, $E_i$ is the emissions rate, $S_i$ is the loss rate, $R_i^{gas}$ is the change in concentration due to gas-phase reactions, $R_i^{part}$ is the change in concentration due to particle-phase reactions and $R_i^{phase}$ is the change in concentration

due to phase change (Held et al., 2005). Loss rates include both dry and wet deposition. Phase change for inorganic species occurs using a kinetic treatment for gas-particle conversion (Hu et al., 2008) driven towards the point of thermodynamic equilibrium (Nenes et al., 1998). Phase change for organic species is also treated as a kinetic process with vapor pressures of semi-volatile organics calculated using the 2-product model (Carlton et al., 2010). More sophisticated approaches for secondary organic aerosol (SOA) formation (Cappa et al., 2016) were also tested in the current study but these required a larger number of assumptions and they did not produce higher SOA concentrations in the $PM_{0.1}$ size fraction.

Nucleation was included in the model using the ternary nucleation (TN) mechanism involving $H_2SO_4$-$H_2O$-$NH_3$ (Napari et al., 2002). A tunable nucleation parameter equal to $10^{-5}$ was used based on results from previous studies across California for the year 2012 (Yu et al., 2018) . Yu et al found good agreement between predicted and measured concentrations of daily-averaged $PM_{0.1}$ and $N_7$ source contributions in California (Yu et al., 2018). The current study expands these nucleation calculations to investigate new particle formation across all major U.S. cities, but the data needed to evaluate the accuracy of these calculations is generally not available outside of California and particle number concentrations will not be a focal point of this work. The model spatial resolution was 4km over the 4.2 million $km^2$ of simulated urban areas and so near-roadway concentrations of ultrafine particles on spatial scales of ~0.1 km also will not be presented.

A total of 50 particle-phase chemical species are included in each of 15 discrete particle size bins that range from 0.01-10 μm particle diameter (Held et al., 2005). Artificial source tags are used to quantify source contributions to the primary particle mass and the secondary organic aerosol (SOA) mass for a specific bin size, therefore allowing for the direct contribution of each source of $PM_{2.5}$ and $PM_{0.1}$ mass to be determined. Gas-phase concentrations of oxides of nitrogen (NOx), volatile organic compounds (VOCs), oxidants, $O_3$ , and semi-volatile reaction products were predicted using the SAPRC-11 chemical mechanism (Carter and Heo, 2013).

## 2.3 Model Inputs

Anthropogenic emissions were generated using the Sparse Matrix Operator Kernel Emissions (SMOKEv3.7) modeling system applied to the 2011 National Emissions Inventory.

The NEI reports county wide emission totals from all 50 states that are then mapped using spatial surrogates. Temporal profiles are also used to account for variation by time of month, week and day, however the NEI does not account for "no-burn" days that would impact residential wood combustion or precipitation events that would impact paved/unpaved road dust. These default profiles may result in larger model performance bias when comparing predictions to measured values. Emissions from each of the four major source sectors (area, mobile, non-road and point) were tagged to create fifteen (15) different emissions groups: on road diesel, on road gasoline, off road diesel, off road gasoline, biomass, food cooking, natural gas, process heaters, distillate oil, aviation, cement, coal, steel foundries, paper products and all other emissions. Size and composition-resolved source profiles were then assigned to the PM emissions within each of these groups using the UCD/CIT emissions processor based on the most recent measurements available in the literature (Robert et al., 2007a;Robert et al., 2007b;Kleeman et al., 2008). Some of the fifteen (15) source categories were represented using weighted average source profiles from multiple sources as described in the supporting information Table S1.

Daily values for 2010 wildfire emissions were generated using the Global Fire Emissions Database (GFED) (Giglio et al., 2013). Biogenic emission rates were generated using the Model of Emissions of Gases and Aerosols from Nature (MEGANv2.1). The gridded geo-referenced emission factors and land cover variables required for MEGAN calculations were created using the MEGANv2.1 pre-processor tool and the ESRI_GRID leaf area index and plant functional type files available at the Community Data Portal (Guenther et al., 2012).

Meteorology parameters used to drive the UCD/CIT CTM and the MEGANv2.1 biogenic emissions were generated using the Weather Research and Forecasting model (WRFv3.6) and WRF preprocessing system (WPSv3.6). Meteorological fields were created within 3 nested domains with horizontal resolutions of 36km, 12km, and 4km, respectively. Each domain had 31 telescoping vertical levels up to a top height of 12km. Four-dimensional data assimilation (FDDA) or "FDDA nudging" was used to anchor meteorological predictions to measured values (Hu et al., 2010). Meteorological data and gridded map projections needed for 2010 emissions modeling were taken from the corresponding WRF simulations using the meteorology-chemistry interface processor (MCIP).

### 2.4 Supporting Measurements

Ambient hourly $O_3$ measurements and daily $PM_{2.5}$ measurements were obtained from the EPA AQS API / Query AirData (EPA). Model predictions are compared to these measurements to build confidence in the accuracy of the overall modeling system since $PM_{0.1}$ measurements are not available during any of the peak summer pollution events studied here

### 3. Results

Predicted maximum 1-hr $O_3$, $NO_2$, $SO_2$ and CO concentrations were compared to measurements at all available monitors within each study CBSA to indirectly evaluate the accuracy of the emissions inventories and meteorology fields. Many of the sources that emit $O_3$ precursors also emit ultrafine particles. Likewise, meteorological parameters including wind speed and mixing depth influence the concentrations of all pollutants including ultrafine particles. Successful prediction of gas phase species is therefore a necessary step in the accurate prediction of ultrafine particle concentrations during summer photochemical smog episodes. Predicted 24-hr $PM_{2.5}$ concentrations were also compared to measurements at all available monitors within each study CBSA. Many of the combustion sources that emit primary particles within the $PM_{2.5}$ size fraction also emit primary $PM_{0.1}$ and/or precursor gases that can condense into the $PM_{0.1}$ size range. The Chemical Speciation Monitoring Network (CSN) operated by the U.S. Environmental Protection Agency (EPA) measures $PM_{2.5}$ mass and chemical composition at more than 260 sites throughout the U.S. including many of the 39 cities studied in the current analysis (Solomon et al., 2014). Full monitor information including latitude, longitude and total number of available measurements for comparison within the simulation period are show in the supporting information Tables S2 – S6.

Figure 3 illustrates the normalized mean bias (NMB) and normalized mean error (NME) for predicted 1-hr maximum $O_3$ against measured 1-hr maximum values for each monitor within a specific modeling domain. Figure S1 in the supporting information illustrates the fractional bias (FB) and fractional error (FE) for predicted 1-hr maximum CO, $NO_2$ and $SO_2$ against measured 1-hr maximum values. Figure 4 illustrates the NMB and NME for 24-hr average predicted $PM_{2.5}$ concentrations against measured 24-hr average $PM_{2.5}$ concentrations at each available monitor over the specific simulation period. A time series of predicted vs. measured $O_3$ concentrations is displayed in Figure S2.

Table 2 summarize the total number of available monitors for comparison of measured values vs. predicted values for $O_3$ and $PM_{2.5}$. Emery et al. (2017) recommend model performance criteria for 1-hr $O_3$ NMB less than or equal to ±0.15 and NME less than or equal to 0.30. 24-hr $PM_{2.5}$ model performance recommendations also based on Emery et al. (2017) are NMB less than or equal to ±0.30 and NME less than or equal to 0.50 (Emery et al., 2017). Table 2 displays the percentage of measured vs. predicted comparisons that met the performance criteria for NME over the entire U.S. modeling domain. In summary, 95% of all locations met NME performance criteria for $O_3$ predictions and 85% of all locations met NME performance criteria for $PM_{2.5}$ predictions.

Elemental carbon (EC) and organic carbon (OC) are the chemical components most relevant for both the $PM_{2.5}$ and the $PM_{0.1}$ size fractions. Figure 5 illustrates predicted vs. measured 24-hr $PM_{2.5}$ EC and OC concentrations for all 39 cities. Primary organic matter tracked by model calculations is converted to OC by dividing by a factor of 1.2 (Russell, 2003). Secondary organic aerosol tracked by model calculations is converted to OC by dividing by a factor of 1.5. In general, the model slightly under predicts $PM_{2.5}$ EC, OC, and mass with regression slopes ranging from 0.62 for EC to 0.71 for OC. The negative bias in model predictions may stem from the 4km spatial averaging inherent in the calculations vs. the influence of sources closer than 4 km to the measurement site in the urban environment such as highways, restaurants, etc. Model performance statistics for $PM_{2.5}$ predictions are summarized in the supplemental information Table S6.

$PM_{0.1}$ measurements are not available for model evaluation in the 39 cities across the U.S. in 2010 at the core of the current study, but measurements are available in California in the years 2015 and 2016 that can serve to evaluate similar modeling procedures. Yu et al. (2018) compared $PM_{0.1}$ concentrations in Los Angeles, Fresno, East Oakland, and San Pablo, California predicted using the UCD/CIT air quality model to receptor-based source apportionment calculations based on measured concentrations of molecular markers in the ultrafine particle size fraction (Xue et al., 2018). Good agreement was found between predictions for $PM_{0.1}$ concentrations associated with gasoline engines, diesel engines, food cooking, wood burning, and "other sources" from these two independent techniques. Further details of this comparison are provided by (Yu et al., 2018). This evaluation of the modeling procedures builds confidence

in the $PM_{0.1}$ source predictions across the U.S. in the current study, but new measurements would be helpful to fully evaluate model predictions in the future.

Figure 6 illustrates a composite representation of $PM_{2.5}$ and $PM_{0.1}$ mass across the U.S. during the summer pollution episodes listed in Table 1. The spatial plot in Figure 6 is constructed using the intermediate 12km simulation results from multiple simulations stitched together to cover a broader geographical area. Regional $PM_{0.1}$ concentrations reach a maximum value of 5 $\mu g/m^3$ in a few isolated grid cells with wildfires but concentrations generally exceed 2 $\mu g\ m^{-3}$ in major urban regions across the U.S. including Los Angeles, the San Francisco Bay Area, Houston, Miami, and New York. The comparison between $PM_{2.5}$ mass (Figure 6a) and $PM_{0.1}$ mass (Figure 6b) shows that predicted $PM_{0.1}$ spatial gradients are sharper with less regional contributions between "hot spots". Locations in the Midwestern and Eastern U.S. outside of cities with high $PM_{2.5}$ concentrations due to secondary formation (sulfate and secondary organic aerosol) did not have corresponding high concentrations of $PM_{0.1}$. Most major urban centers had noticeable peaks of both $PM_{2.5}$ and $PM_{0.1}$. This pattern presents a challenge for epidemiological studies seeking to differentiate the effects of $PM_{2.5}$ and $PM_{0.1}$ because the locations with differential exposure (high $PM_{2.5}$ but low $PM_{0.1}$) have low population density, which will reduce the power of the analysis.

The UCD/CIT model explicitly tracks source contributions to particle mass in each size bin using artificial source tags. Pie charts of $PM_{2.5}$ and $PM_{0.1}$ source contributions are illustrated in Figure 6 for selected major cities. Pie charts for $PM_{0.1}$ source contributions in all 39 U.S. cities are shown in Figure 7. The detailed source profiles within each city are based on the nested 4km simulation results during the pollution events listed in

Table 1. Source contribution spatial plots for the entire U.S. are shown in the supplemental information Figure S3 through S5 and pie charts for $PM_{2.5}$ source contributions in all 39 U.S. cities are shown in the supplemental information figure S6. On-road gasoline and diesel vehicles made significant contributions to regional $PM_{0.1}$ in all 39 cities even though peak contributions within 0.3 km of the roadway were not resolved by the 4 km grid cells. Food cooking also made significant contributions to $PM_{0.1}$ in all cities but biomass combustion was only important in locations impacted by summer wildfires. Residential wood combustion is not typically a strong source in the summer due to the warmer temperatures however in the winter

time biomass would most likely be a dominant source. Aviation was a significant source of $PM_{0.1}$ in cities that had airports within their urban footprints. Industrial sources including cement manufacturing, process heating, steel foundries, and paper & pulp processing impacted their immediate vicinity but did not significantly contribute to $PM_{0.1}$ concentrations in any of the target 39 cities. Natural gas combustion made significant contributions to $PM_{0.1}$ concentrations due to the widespread use of this fuel for residential, commercial, and industrial applications. Natural gas contributions were especially significant in locations with high levels of industrial use such as chemical refineries or in locations with significant levels of natural gas fired power plants.

The major sources of primary $PM_{0.1}$ and $PM_{2.5}$ were notably different in many cities (compare Figure 6a and Figure 6b). The sources that contribute most strongly to $PM_{2.5}$ are on road diesel, gasoline, food cooking, coal and "other" which includes brake and tire wear from mobile sources and dust. Natural gas combustion makes minor contributions to primary $PM_{2.5}$ mass since particles from this source have a mass distribution peaking at ~0.05 μm particle diameter (Chang et al., 2004) with all of the emitted mass in the $PM_{0.1}$ size fraction. In contrast, other combustion sources using more complex fuels such as on-road vehicles have a mass distribution peaking at ~0.1 μm with at least half the emitted mass outside the $PM_{0.1}$ size fraction (Robert et al., 2007b;Robert et al., 2007a). Likewise, food cooking contributes strongly to $PM_{2.5}$ concentrations but the emitted particle mass distribution peaks at 0.2 μm with the majority of the mass outside the $PM_{0.1}$ size fraction.

The fraction of PM that is primary within each CBSA is listed in Supporting Information Tables S7-16. Averaged across the U.S., $PM_{2/5}$ was found to be approximately 62% primary material while $PM_{0.1}$ was found to be approximately 87% primary material.

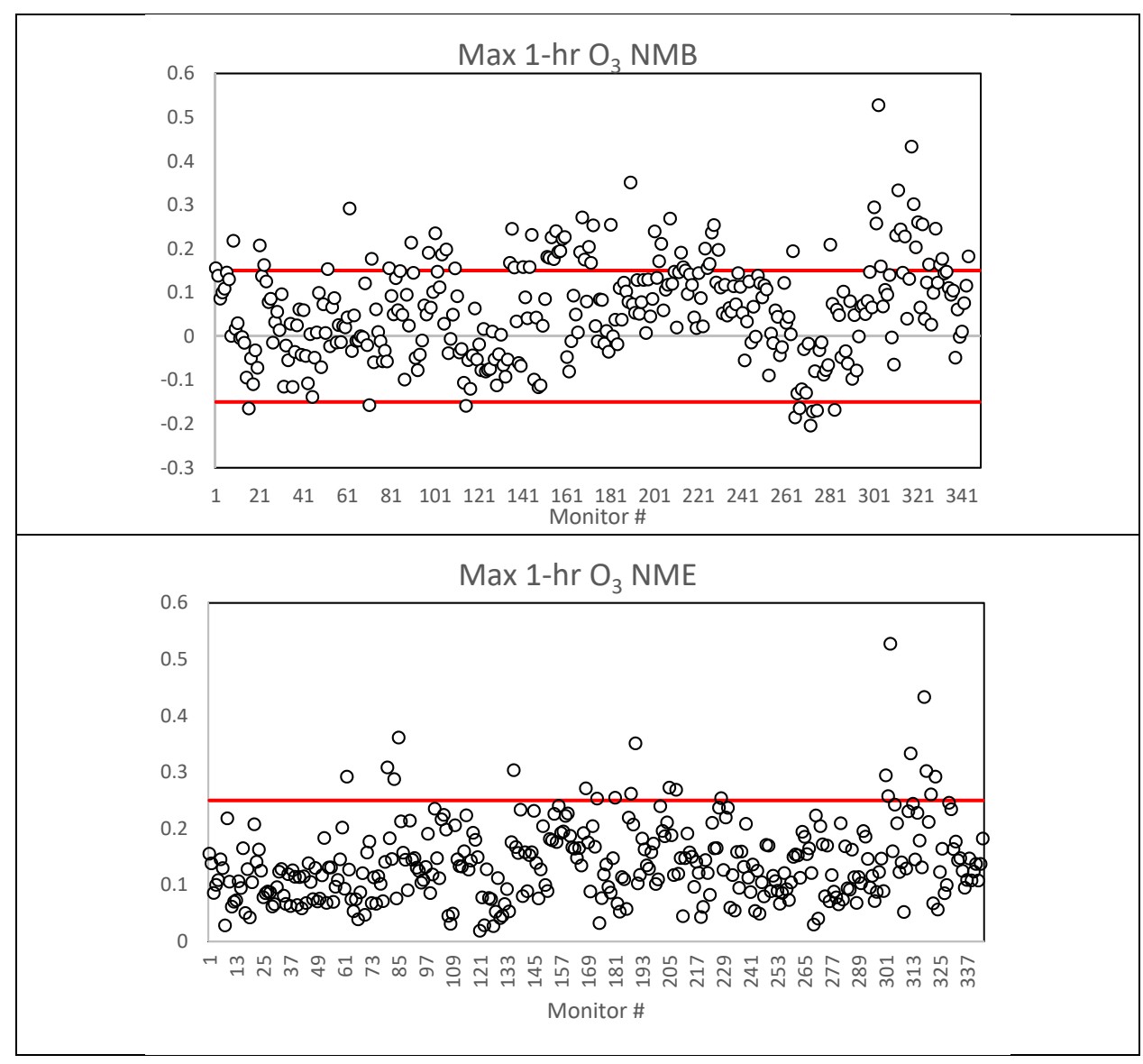

Figure 3 Model performance statistics for predicted maximum 1-hr $O_3$ against measured values. Red line represents performance criteria of 0.15 for NMB and 0.25 for NME. NMB and NME were calculated for available measurements against predictions at every monitor in the CBSA region based of the U.S. EPA AQ datamart. Monitor # (horizontal axis), latitude and longitude, name, MO, MP, NMB, NME, FB and FE value available for all monitors in supporting information

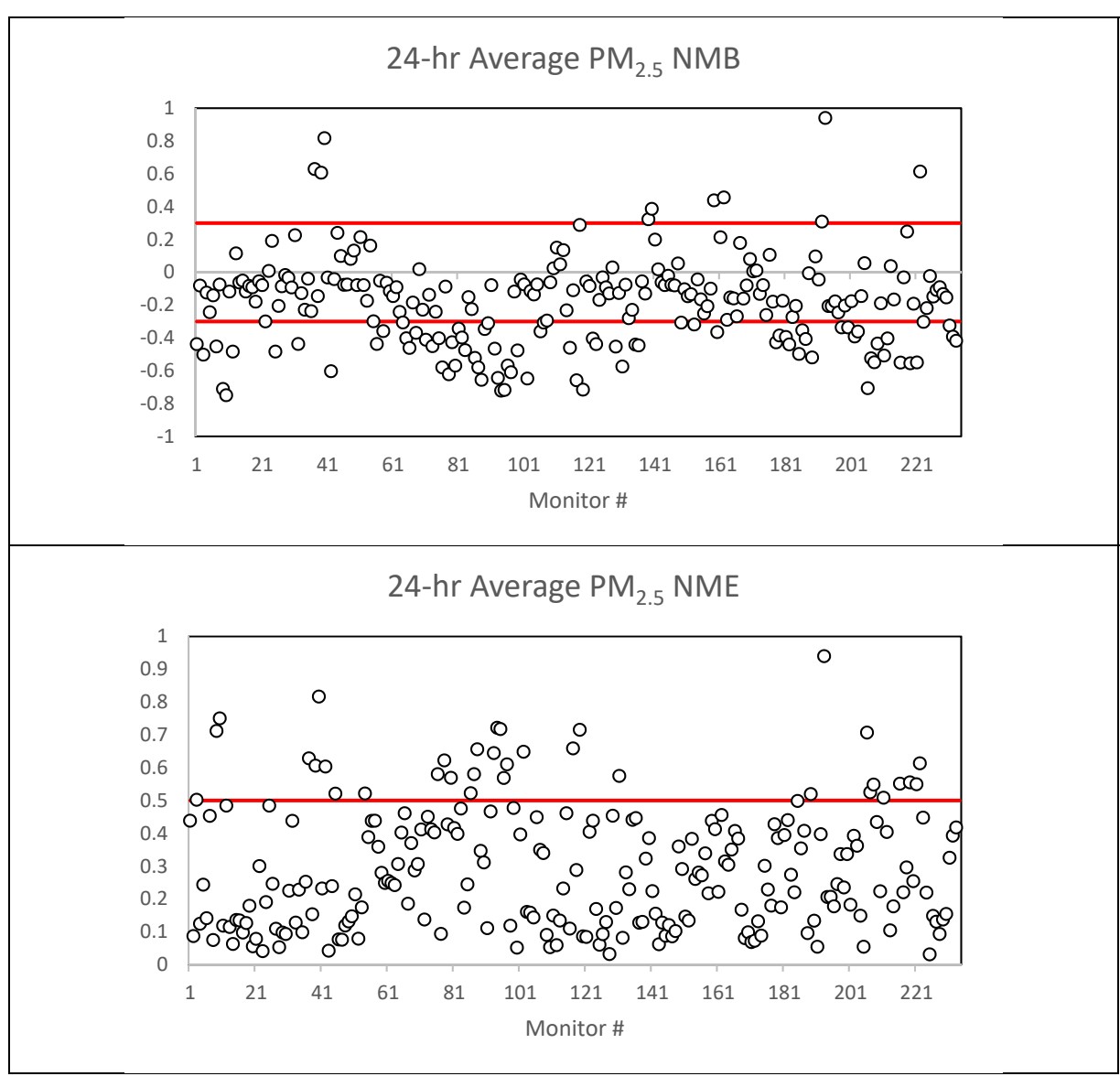

Figure 4 Model performance statistics for predicted 24-hr average PM2.5 against measured values. Red line represents performance criteria of +/- 0.30 for NMB and 0.50 for NME. Normalized mean bias and normalized mean error were calculated for available measurements against predictions at every monitor in the CBSA region based of the U.S. EPA AQ DataMart. Monitor # (horizontal axis), latitude and longitude, name, MO, MP, NMB, NME, FB and FE value available for all species in supporting information

Table 2 Percent of Monitors throughout entire U.S. domain that met performance criteria for normalized mean error (NME)

| Species | Total Number of Monitors within 4km modeling domain for Comparison to Predicted Values | % that met NME performance Criteria |
|---|---|---|
| Max 1-hr $O_3$ | 344 | 95% |
| Average 24-hr $PM_{2.5}$ | 234 | 85% |

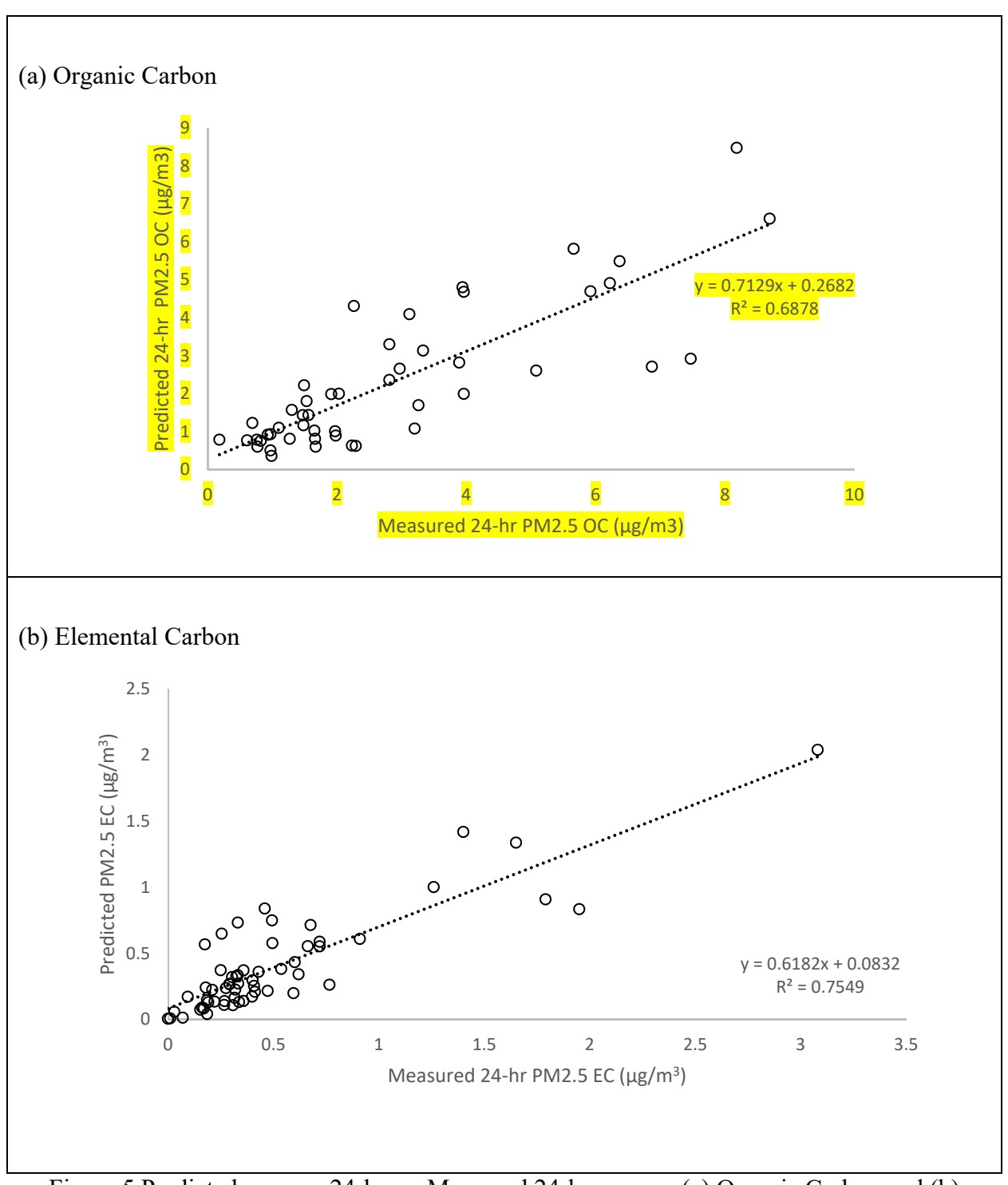

Figure 5 Predicted average 24-hr vs. Measured 24-hr average (a) Organic Carbon and (b) Elemental Carbon ($\mu g/m^3$). Predicted OC was converted from predicted organic matter (OM) and secondary organic aerosol components using a ratio of 1.2 and 1.5, respectively (Russell 2003).

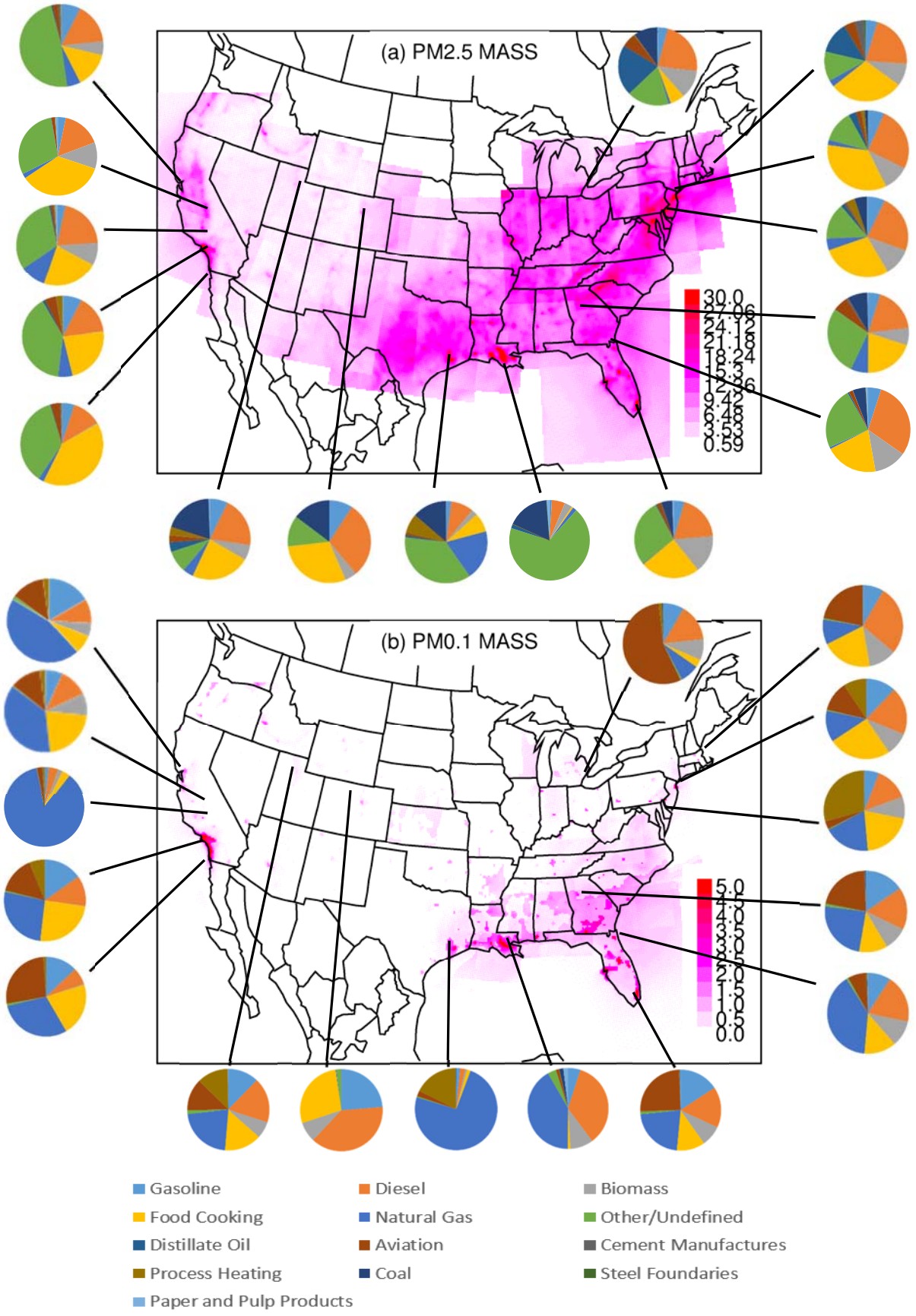

Figure 6 (a) $PM_{2.5}$ and (b) $PM_{0.1}$ 24-hr average mass ($\mu g/m^3$) during summer air pollution event. Scale drawn to highlight all areas of US. Actual Max for (a) = 94.25 $\mu g/m^3$ (b) = 9.43 $\mu g/m^3$.

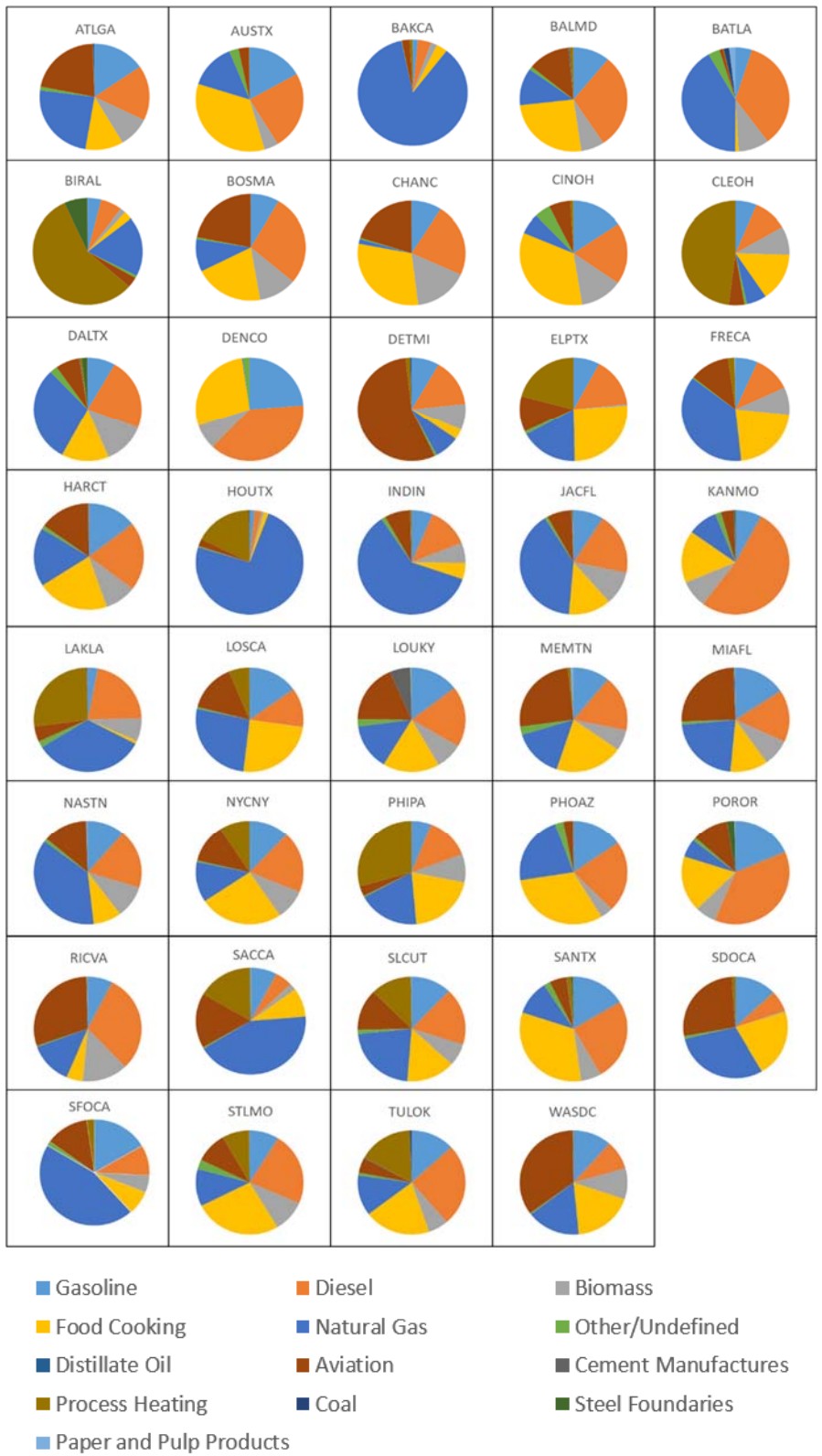

Figure 7 PM$_{0.1}$ source contribution for 39 cities across the continental US

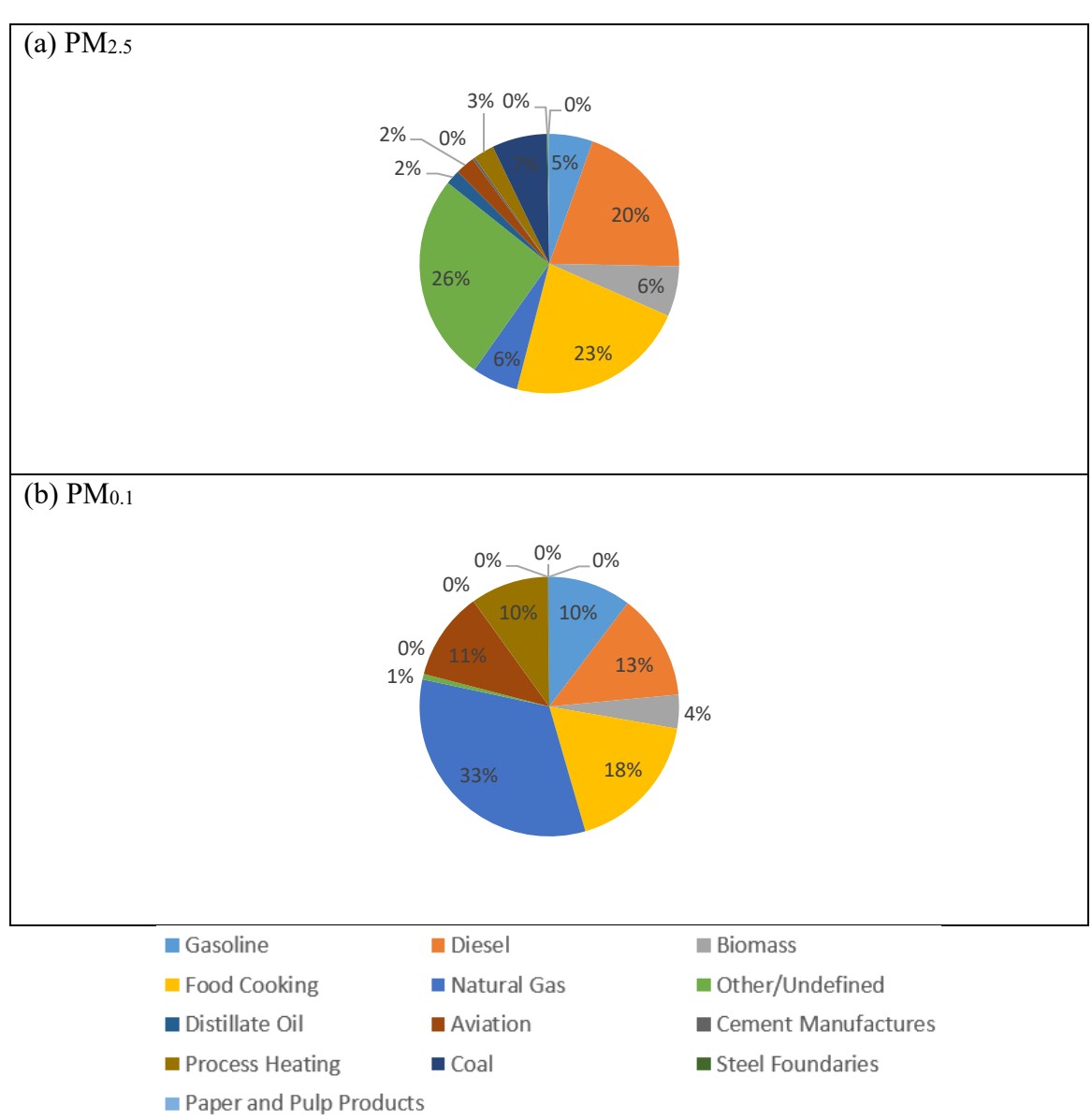

Figure 8 Population weighted average source contribution across the 39 major cities in the continental U.S. for (a) PM$_{2.5}$ and (b) PM$_{0.1}$

## 4. Discussion

Figure 8 illustrates the population-weighted average $PM_{0.1}$ source contributions across all 39 study cities shown in

Table 1. These predictions are based on source profile measurements for wood burning, food cooking, mobile source and non-residential natural gas combustion reported in multiple peer-review studies (Taback et al., 1979;Cooper, 1989;Houck et al., 1989;Hildemann et al., 1991b;Hildemann et al., 1991a;Harley et al., 1992;Schauer et al., 1999a, 1999b, 2001, 2002b, 2002a;Kleeman et al., 2008;Kleeman et al., 2000;Robert et al., 2007b;Robert et al., 2007a). In addition, new measurements made by Xue et al. (2019) were conducted to confirm previous measurements of the particle size distribution associated with natural gas and biomethane combustion particles.

The results summarized in Figure 8 highlight the importance of natural gas combustion particles in the $PM_{0.1}$ size fraction and the minor role that these natural gas combustion particles play in the $PM_{2.5}$ size fraction. Natural gas typically consists of +93% methane with the balance of the fuel made up by higher molecular weight alkanes and trace impurities. In addition to background sulfur compounds in the natural gas, sulfur-containing odorants such as mercaptans are commonly added to aid in leak detection. Natural gas combustion does not emit high amounts of particulate matter per J of energy in the fuel, but the widespread use of natural gas suggests that it could still contribute significantly to ambient $PM_{0.1}$ concentrations. Natural gas combustion accounted for 29% of total U.S. energy consumption in 2016 (U.S. Department of Energy, 2017). In contrast, gasoline combustion accounted for 17% of U.S. energy consumption and diesel fuel combustion accounted for approximately 6% of U.S. energy consumption in 2016. Less than half of the PM emitted by gasoline and diesel fuel combustion is in the $PM_{0.1}$ size fraction (Robert et al., 2007a;Robert et al., 2007b) whereas all of the PM emitted by natural gas combustion is in the $PM_{0.1}$ size fraction (Chang et al., 2004). Taken together, these facts support the potential importance of natural gas combustion for ambient $PM_{0.1}$ concentrations.

The five (5) states with the highest consumption of natural gas in 2016 were Texas (14.7%), California (7.9%), Louisiana (5.7%), New York (5%), and Florida (4.8%). These consumption patterns are reflected in the natural gas distribution system (Figure 9a) and the predicted $PM_{0.1}$ concentration field associated with natural gas combustion (Figure 9b). Natural

gas end-use included electric power generation (36%), industrial applications (34%), residential use (16%), commercial use (11%), and transportation (3%).

Lane et al. (2007) used a source-resolved version of PMCAMx and individual emission inventories to determine source contributions of primary organic material ($POM_{2.5}$) (Lane et al., 2007). Lane et al. note that $POM_{2.5}$ associated with natural gas sources ranged from 0.1 to 0.8 $\mu g/m^3$. Chang et al in 2004 measured emitted particle size distributions for gas-fired stationary combustion that fell between 10-100nm (Chang et al., 2004). The combination of these two results indicates that the natural gas mass component of $POM_{2.5}$ predicted by Lane et al. is consistent with the magnitude of the $PM_{0.1}$ mass associated with natural gas combustion found in the current study. Lane et al. were not studying $PM_{0.1}$ and so the major role of natural gas in this size fraction was not identified.

Posner and Pandis (2015) utilized PMCAMx with the LADCO 2001 BaseE source-resolved mass emissions inventory for a July 2001 prediction of $N_x$ over the Eastern U.S. with 36 km resolution (Posner and Pandis, 2015). Posner and Pandis used a "zero-out" method in combination with source-specific size distribution to study the percent contribution of six major sources (on road gasoline, industrial, non-road diesel, on road diesel, biomass and dust) of $N_x$. They found that $N_x$ was made up of 36% on-road gasoline, 31% industrial, 18% non-road diesel, 10% on-road diesel, 1% biomass burning and 4% long-range transport (Posner and Pandis, 2015). The emissions particle number inventory was normalized based on $PM_{10}$ mass from each source and particle emissions from natural gas sources were assumed negligible, which effectively removed natural gas sources from the simulation. This has minor effects on $PM_{2.5}$ and $PM_{10}$ predictions, but the results of the current study suggest that natural gas combustion contributions significantly to ultrafine particle concentrations.

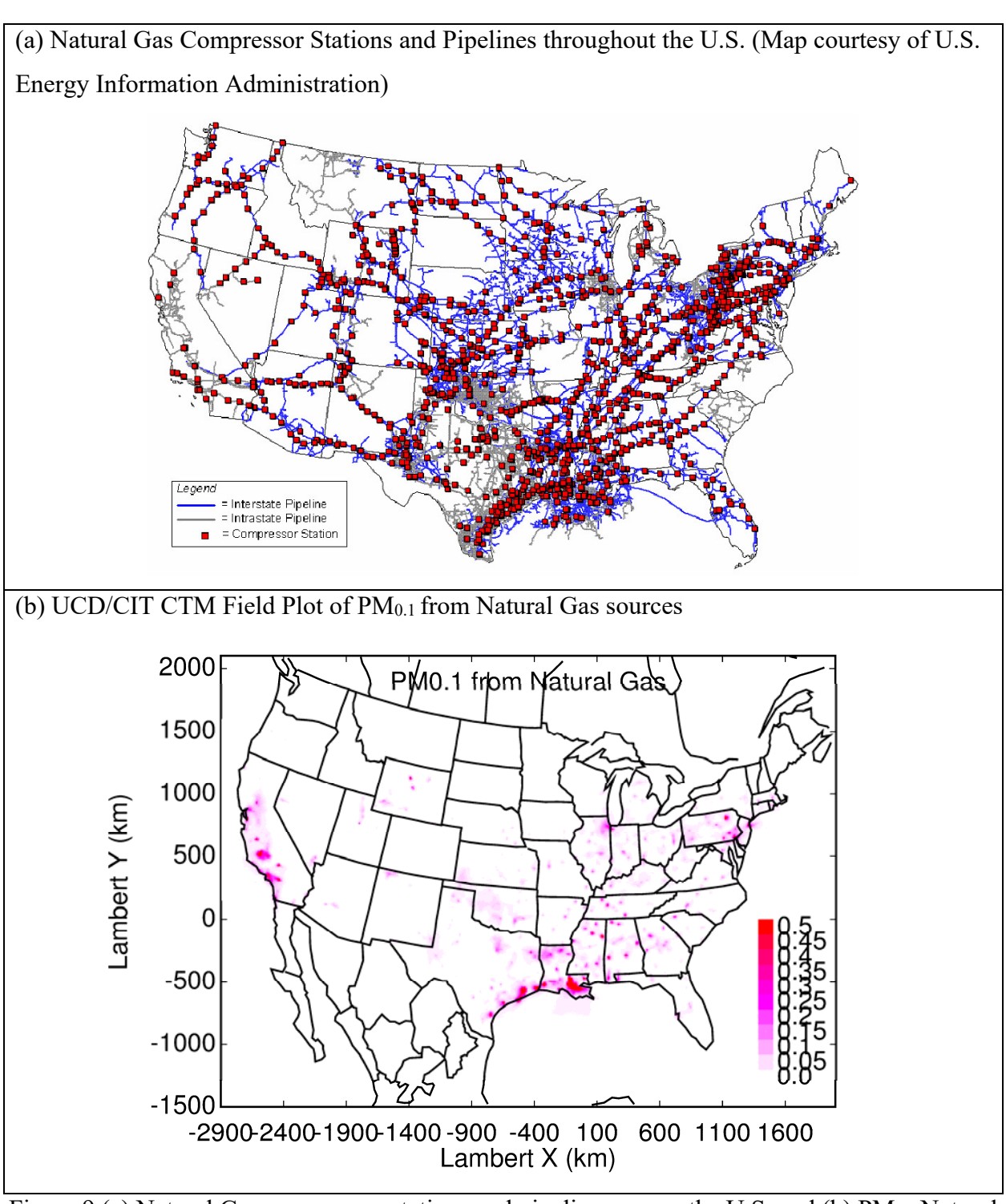

(a) Natural Gas Compressor Stations and Pipelines throughout the U.S. (Map courtesy of U.S. Energy Information Administration)

(b) UCD/CIT CTM Field Plot of $PM_{0.1}$ from Natural Gas sources

Figure 9 (a) Natural Gas compressor stations and pipelines across the U.S. and (b) $PM_{0.1}$ Natural Gas combustion concentrations ($\mu g/m^3$).

Future epidemiological studies may be able to differentiate $PM_{0.1}$ and $PM_{2.5}$ health effects by contrasting cities with different predicted ratios of $PM_{0.1}$ / $PM_{2.5}$. Although the current study does not calculate the annual-average concentrations that would be needed for such an analysis, the results for the peak photochemical episodes may provide some useful insights to guide future studies. Figure 10 illustrates the correlation between predicted $PM_{2.5}$ and $PM_{0.1}$ concentrations in the 39 cities considered in the current analysis, Figure 11 illustrates the ratio of $PM_{0.1}/PM_{2.5}$ for each city, and Figure 12 illustrates a field plot showing the ratio of $PM_{0.1}/PM_{2.5}$ across the continental US. Cities with higher $PM_{0.1}$ / $PM_{2.5}$ ratios include Houston TX, Los Angeles CA, Salt Lake City UT , Cleveland OH, and Bakersfield CA. Cities with lower $PM_{0.1}$ to $PM_{2.5}$ ratios include Lake Charles LA, Baton Rouge LA, St. Louis MO, Baltimore MD, and Washington DC. Measurements should be conducted in these locations to verify the contrast in $PM_{0.1}$ / $PM_{2.5}$ concentrations in preparation for future exposure analysis.

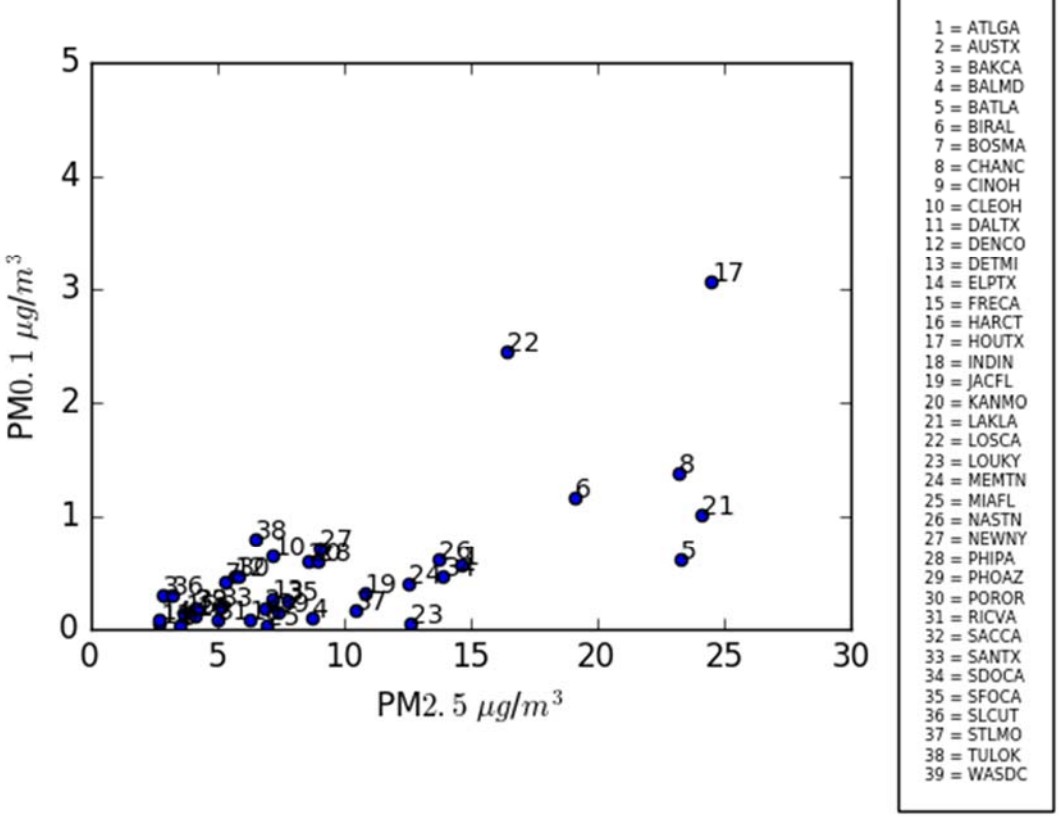

Figure 10 Scatter plot showing correlation between 24-hr average $PM_{2.5}$ and $PM_{0.1}$ for the 39-cities.

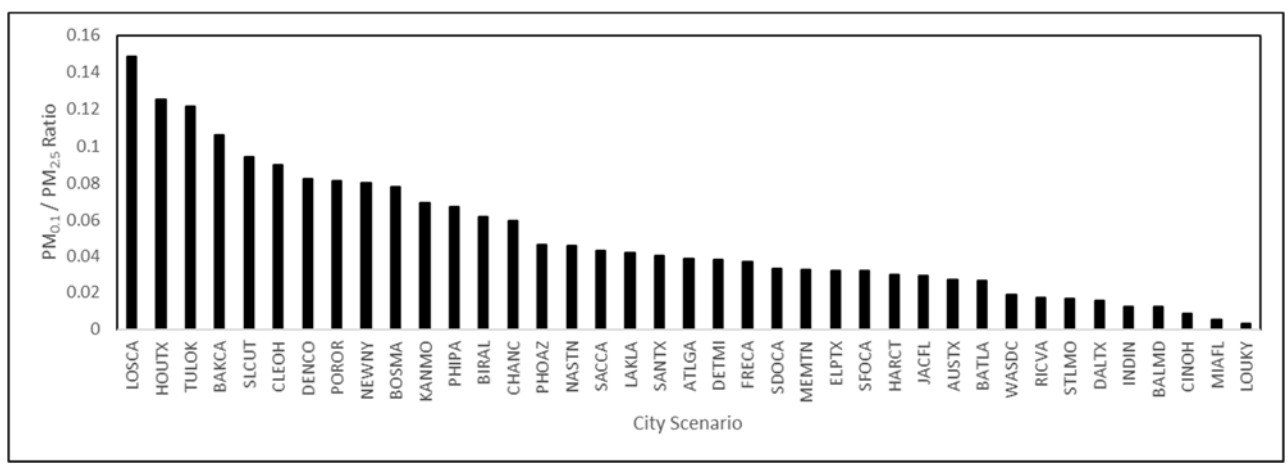

Figure 11 PM$_{0.1}$/PM$_{2.5}$ ratio for each city

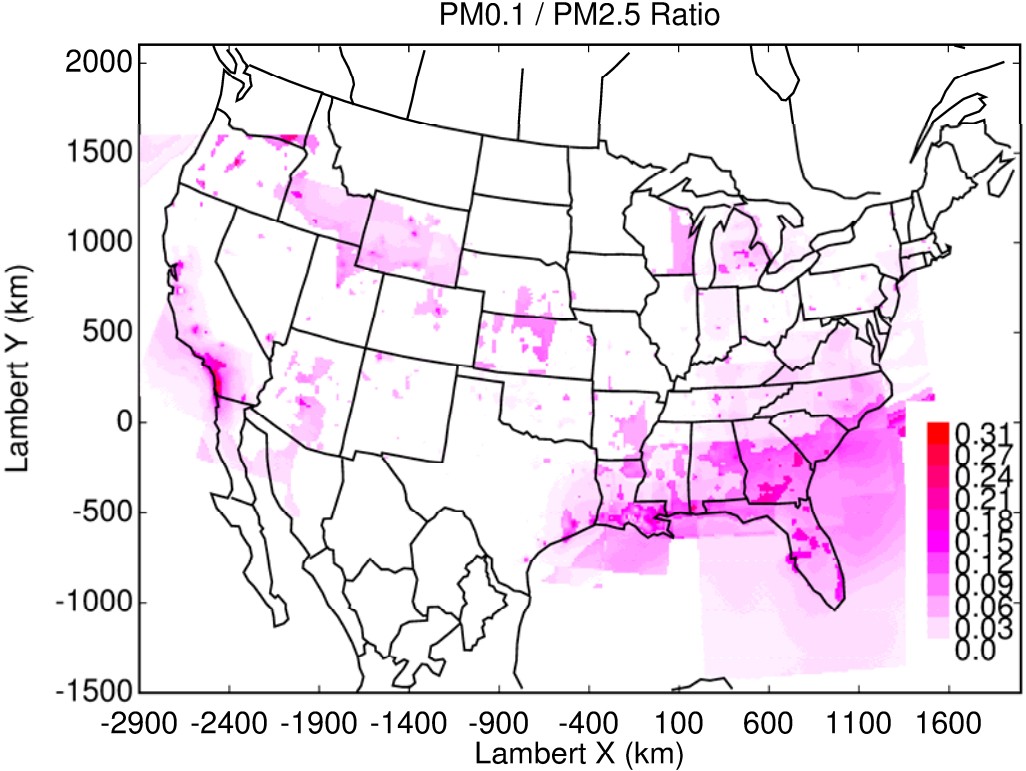

Figure 12 PM$_{0.1}$/PM$_{2.5}$ ratio across the U.S.

## 5. Conclusion

The UCD/CIT regional chemical transport model was used to predict source contributions to $PM_{0.1}$ across the continental U.S. during peak photochemical smog periods during the year 2010. Performance for $PM_{2.5}$ and $O_3$ predictions met or exceeded the criteria typically used for regional air quality model applications building confidence in the emissions inputs and meteorological fields used to drive the calculations. Similar model exercises carried out for episodes in California in 2015 and 2016 find good agreement between predicted $PM_{0.1}$ source contributions and receptor-based $PM_{0.1}$ source contributions calculated using measured concentrations of molecular markers (Yu et al., 2018). In the current study, predicted regional $PM_{0.1}$ concentrations exceeded 2 $\mu g\ m^{-3}$ during summer pollution episodes in major urban regions across the U.S. including Los Angeles, the San Francisco Bay Area, Houston, Miami, and New York. Predicted $PM_{0.1}$ spatial gradients were sharper than predicted $PM_{2.5}$ spatial gradients due to the dominance of primary aerosol in $PM_{0.1}$. This finding suggests that $PM_{0.1}$ measurement networks needed to support epidemiology must be denser than comparable $PM_{2.5}$ measurement networks. Non-residential natural gas combustion was identified as a major source of $PM_{0.1}$ across all major cities in the U.S. On-road gasoline and diesel vehicles contributed on average 14% to regional $PM_{0.1}$ even though peak contributions within 0.3 km of the roadway were not resolved by the 4 km grid cells. This is consistent with other studies that have found an exponential decrease in ultrafine particle concentrations downwind of major roadways (Wang et al., 2011) due to the sharp gradient of $PM_{0.1}$. Food cooking also made significant contributions to $PM_{0.1}$ in all cities but biomass combustion was only important in locations impacted by summer wildfires. Aviation was a significant source of $PM_{0.1}$ in cities that had airports within their urban footprints. The major sources of primary $PM_{0.1}$ and $PM_{2.5}$ were notably different in many cities. Future epidemiological studies may be able to differentiate $PM_{0.1}$ and $PM_{2.5}$ health effects by contrasting cities with different ratios of $PM_{0.1}$ / $PM_{2.5}$ sources.

Data Availability: All of the $PM_{0.1}$ and $N_x$ outdoor exposure fields produced in the current study are available free of charge at http://faculty.engineering.ucdavis.edu/kleeman/. The model source code and input data are available to collaborators through direct email request to the corresponding author.

## 6. Author Contributions

M. Venecek prepared model input data, performed model simulations, postprocessed model output, and prepared the initial draft of the manuscript. X. Yu created the nucleation module used in model calculations. M. Kleeman designed the study, created the models used for the calculations, assisted in model simulations, assisted in postprocessing model output, and revised the final manuscript.

## 7. Acknowledgements

This research was supported by the California Air Resources Board under project #14-314. Neither CARB nor any person acting on their behalf: (1) makes any warranty, express or implied, with respect to the use of any information, apparatus, method, or process disclosed in this report, or (2) assumes any liabilities with respect to use, or damages resulting from the use or inability to use, any information, apparatus, method, or process disclosed in this report

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
