# Peer review of "Ultrafine Particulate Matter Source Contributions across the Continental United States"

_Atmospheric Chemistry and Physics, 2018_

## Referee Comment (RC1) · Anonymous Referee #2 · 28 Sep 2018

This manuscript focuses on the simulation of ultrafine particle mass during summer pollution episodes across the United States. Given the recent interest in these smaller particles due to their impact on the health of exposed populations, I find the topic to be relevant to ACP. The paper is generally well written (see comments below), making it straightforward to follow. Tables and Figures are appropriate, as are the citations, abstract, and conclusions. The methodology used is proven and scientifically sound. Based on these, assuming the issues below are addressed, I recommend publication.

1. Page 2, lines 22-23. I would suggest removing "As expected." This minimizes the work, as if it were expected, why bother?

[Figure]

2. In general, the paper could deal with some cleaning up of language, punctuation, etc. Examples Page 2, lines 31-32, use . . . for commercial use Page 3, line 57, should this be low birth weight Page 3, line 64, word national is not necessary, as monitors in the continental US are specified Page 4, lines 80 and 100 (and elsewhere) US or U.S.? On page 22, line 410, United States is written out. In the SI, page 14, line 118, states is not capitalized. Page 4, line 85, add 'to' between exposure and ultrafine Page 7, line 156, missing a closing parenthesis Page 9, line 194 (and elsewhere including Fig 1 and Fig S2 caption), vs. not vs Page 17, lines 337-340 – Chang et al. (2004) measured. . . Add (2007) to Lane et al. In the SI, Figure S3 should appear after Table S2, as it is cited after Table S2 in the main text. SI, page 14, line 111, Figures compare, not compares

3. In the abstract (Page 2, lines 35-37) and on page 20, lines 364-366), 'higher' and 'lower' ratios should be quantified. Is there a cutoff to determine higher versus lower based on the scatter plot shown in Figure 7?

4. Page 3, line 60. I assume this should be surface area to volume ratio, not just surface area?

5. Page 5, Table 1. Please provide more information about why these 39 cities were selected. Was it the availability of observations? Was it the number of O3 days above 70 ppb? As an example, why Charlotte and not Raleigh, NC? Or why Tulsa and not Oklahoma City, OK? Why were Pittsburgh and Chicago not included?

6. Page 7, line 141. Please justify why nucleation is not considered. This is in line with a later comment about fraction of PM0.1 that is secondary versus primary.

7. Section 2.3 and page 8, line 188. Please provide more information about monitors used and the comparison to model output. It says 'measurements averaged' – does this mean multiple monitors were used? Or was a single monitor compared to the model output for the grid cell in which it resides? For cities with multiple monitors with a grid cell/domain, if multiple are used, it would be appropriate to include that information

(perhaps in the SI).

8. Page 8, line 190. Even though it appears that secondary material is not a huge contribution to PM0.1, it would be appropriate to say 'also emit ultrafine particles and their precursors'

9. Page 9. In reviewing Table S2, it appears that only one city does not meet the MFE for O3? If that is the case, it should be more specific on lines 199-200. It would be appropriate to provide the average O3 model performance statistics at this point. Then at the bottom of the page, the authors could discuss PM model performance statistics (and again, specify that only one city does not meet the MFE for PM model performance). Right now, it is slightly confusing to discuss O3, then PM, then both in terms of the averages.

Page 10, line 231. While I recognize that the submitted Yu et al. manuscript describes the 'good agreement' for PM0.1 modeling assessment in California, I think it could be summarized more quantitatively here in only one or two sentences.

Page 11, line 262. I recognize that the focus of this work is summer. However, would it be appropriate to highlight that the biomass contribution might be different in winter when wood burning for home heating could be a prevalent source of PM0.1 in colder regions?

Page 13, Figure 2. How does the model convert from OM to OC? Does the two product model used (Carlton et al.) predict OM or OC? I thought it was OM, but if I am mistaken please correct me. If a conversion is done to estimate OC based on the simulated OM, it would be appropriate to include this in the caption to Figure 2.

Page 14, Figure 3; Page 15, Figure 4. Would it be possible to somehow show on these figures the relative contribution of primary PM0.1 versus secondary PM0.1? This would truly drive home (and quantify) the relative contributions of direct emission versus in situ formation (I realize it is predominantly primary, but doing this would show it).

**[ACPD](ACPD)**

Interactive
comment

Page 18, line 343. This paragraph does not seem necessary to me, as it focuses on previous work that simulated PNC, which as the authors point out in the nucleation discussion (see comment above), is not equivalent to PM0.1 (the focus of this work).

Page 22, Figure 8. A suggestion for improved readability: break up this figure into four panels by geographic region of the nation (since the focus is determining how cities in the same region compare – as discussed as 'regional clusters' on page 21).

SI, Page 13, line 95. The MFE given in the caption (0.67) is for O3? The MFE for PM given in the text is 0.75. Please specify both in the caption here. Also, note that the bold lines reflect cities that do not meet one of those criteria.

SI, Tables S3 and S4. These do not appear to be called out anywhere (if they were, and I missed it, I apologize). I assume this is the data that were used to create the vectors for the dot products? If so, that discussion is an appropriate place for them to be called out.

---

## Referee Comment (RC2) · Anonymous Referee #1 · 6 Oct 2018

Review of Veneek et al., "Ultrafine Particulate Matter Source Contriutions across the Contentnal United States

In this paper, Veneek et al., have applied the UCD model to simulate PM2.5 in 39 different cities in the United States to identify source contributions. The paper addresses an interesting and potentially important area, i.e., what are the sources of ultrafine particles. A similar paper was recently published in ACP by the Pandis group. First, just to be more clear, the title of the article should include "simulated" or "modeled". What makes this article rather unique is the great lack of explanation of many important aspects. First, they choose 39 cities? What was the basis for the choice of cities? They

appear to be most of the large cities, but it would still be good to know why those were chosen and others were not (fundamentally, did they choose the largest 39 cities, and if not, why were others excluded? Having a research rational is important.). The next very, very important issue is that they chose very limited time chunks in each of the cities, some overlapping, but a very odd collection. What motivated such a choice? If it was model-performance driven, than any statement about good model performance is not so relevant, or at least should be taken very carefully and explained, as the model performance metrics have not been designed for allowing the ability to choose specific locations and times when the model is performing adequately. (This would be akin to letting epidemiologists to go back and choose periods that specifically do or do not find associations. That would be viewed as bad practice and not acceptable.) It is my understanding, and I have checked this with colleagues, that the typical modeling with a state-of-the-science model like CAMx (which is also a regional chemical transport model, similar to UCD), it is applied over the domain over a chosen period, and then you look at all of the results for a model evaluation. (When applied in SIPS, both the location and time period are predetermined.) Thus, what is appears is done here, but not stated, is that it was applied over the continental US (or some similar domain) for some period(s), again not explicitly stated. The manuscript should state what was the actual modeling period used, or did they choose the specific simulation dates a priori for each city and just simulate those? This must be stated if it is the case. If that is indeed, the case, was the model started on the beginning date chosen, or was there a period allowed for the initial conditions to be minimized? Also, might the authors better justify choosing episodes beginning in March or Octdober. The former seems a bit early, the latter a bit late. Were those the peak episodes that year? Also, the very limited time periods chosen limit the importance of the manuscript. Are we to take a 3-4 day period of the year to represent the source impacts for the whole year? I would expect a rather different set of sources in the winter than the summer. This whole area is not explored or discussed. I would have thought that that the choice might be driven by the availability of observational data. I realize ultrafine observations are rare, but

it would have been preferred to choose time periods when they were available. Why choose 2010 if UFP measurements are available other years? They might look at the recent Pandis paper (https://www.atmos-chem-phys.net/18/13639/2018/) doing a similar exercise over Europe. The choice of just peak ozone events may very well bias the assessment of source impacts on PM0.1 and PM2.5. Why not choose peak PM2.5 events? That would seem more natural. The model performance part is also rather opaque and requires references. First the model performance should be brought in to the main manuscript, e.g., as done in most modeling papers, showing overall performance across the entire domain and modeling period (not just selected locations and periods). See Simon et al., (2014) for the metrics typically provided and a more complete discussion of model performance evaluation. The working of the paragraph beginning on line 220 is also rather strange, it says ""...MFB values lower than 0.15 and MFE values lower than 0.35 are considered the goal or "excellent" in model performance." Then they go on to say they do not meet them. First, I don't think I have seen EPA have "excellent" as a performance description associated with those levels. I was looking for a citation here (a citation to the specific EPA evaluation metrics is required, as well as adherence to the terminologies used. My understanding is that the current EPA guidance is found at https://www3.epa.gov/ttn/scram/guidance/guide/Draft_O3-PM-RH_Modeling_Guidance-2014.pdf, and looking through that document, I don't see them use the term excellent in terms of performance associated with any metric. (Indeed, a search of that document for the term "excellent" found only two occurrences, one in terms of protocol, another in terms of conceptual model.) That said, if the model does not meet the specific guidance levels, what does it meet? If the guidance was developed for regional scale modeling, without allowing selection by location and time, what does that imply here?). Second, if you do not meet them, what does that mean? Unlike the authors, I took the evaluation as not "building confidence in the accuracy of the model results...", but left me questioning it. I would very much recommend that the authors follow the EPA guidance (or other recent articles, e.g., those by the AQMEII initiative or Ramboll: Emery et al.) in terms of how to conduct,

and report, model performance. Having now looked at the Ramboll study (JAWMA, doi/full/10.1080/10962247.2016.1265027), their ozone and PM performance do not meet the "goal", nor fully meet the recommended levels for "criteria". One could also follow the approaches recommended by the AQMEII initiative. The authors should definitely provide current citations for model performance for which they are using as their benchmarks. Thinking more holistically, the proper approach here would be to apply the model to a whole year, or, if that is computationally infeasible (which should be stated), be applied to one month periods in each season, and the results from each of those months be given. If they only want to consider the peak photochemically-active periods, they should choose a three month period (or more, preferably) that will capture events in all the cities for that year. I was a bit puzzled by the explanation given for Fig. 1. A 4 km grid is pretty fine, and the mobile sources in a typical urban area like LA are pretty ubiquitous. There are a number of monitors in LA: do any of them not go to zero at night? If not, that might suggest a different issue The mismatch in the evening needs a bit more discussion and justification. They commit to providing the outdoor exposure fields. They should also provide the model and its inputs. I assume this is journal policy, but the authors should likewise commit. A main conclusion of the paper is that natural gas is the main contributor to population-weighted exposure. This is a rather unique result and certainly requires more justification and discussion in light of the work that has shown via careful experiments that mobile sources and air craft are major sources of natural gas. Where is the empirical evidence of natural gas being a main contributor and can they show that they have captured the contribution of those other two sources? Their explanation of why Lane et al., or Posner and Pandis is not sufficient to argue that the current results are reliable. How well does the current study capture the spatial dynamics found by the groups from USC (Sioutas), Harvard (Spengler) and UW (e.g., Atmospheric Environment Volume 139, August 2016, Pages 20-29) which tend to point the finger at mobile sources and aircraft, so much so, the latter claim that ultrafine particle counts can be used as a tracer for aircraft turbine emissions. Given the lack of empirical evidence, compounded with their not having done any evaluation of the

ultrafine results against observations nationally, the speculative nature of this section suggests it should be removed, or couched in very different terms (i.e., noting the limitations, with a statement of the speculative nature). The question should be asked if there is sufficient evidence to support controls on a source based on the current analysis. The performance evaluation example (e.g., in reference to the EPA goal/excellence criteria) is not the only place where a citation is needed. They state that a number of other locations have PM0.1 levels above 2 ug/m3 (line 418). Then they go on to say there were sharper gradients in the observations. This would seem to contradict their findings/interpretation. More discussion needed. The entire last paragraph seems to contradict the findings of this paper. Again, they should compare their findings more directly to observations. In the summary, they state their analysis was for "peak photochemical periods during the year 2010." That is a rather strange way to characterize periods during which, in Atlanta, the 8-hr maximum ozone shown reached only about 50 ppb; I visually averaged, the actual value would be useful) in Cincinnati, about 60, in Los Angeles about 90, in New York about 85 ppb. I am not sure about the other locations, but I think the design value for LA in 2010 was about 120. In Cincinnati, it was about 0.079 (https://www3.epa.gov/airquality/greenbook/hbtcw.html), In Atlanta, about 80 ppb. The chosen periods would not appear to be "peak photochemical smog" periods. How different would Fig. 8 be if you simply used the local emissions? It is not apparent what this analysis adds and how it might be useful beyond simply using the inventoried sources. Line 425 uses the word "consistent" after saying that the model could not resolve the observed mobile source peaks. While one can see what they may be trying to say, it should be said differently, and more precisely. One could easily have said, experiments have found peak ultrafine particle levels were tied to on-road mobile sources and aircraft emissions, though those three sources combined account for only 22%, and are thus inconsistent with the results here that identify natural gas combustion. Minor: First line: should be "concentrations" L 156: Missing ")"

4 km stated in both of the first lines of the Abstract.

Fig. 2. Are these the individual, daily values for each site? If so, why are there not more circles? A bit more information in the figure caption would be useful.

Figure 3. What do they mean by "air pollution event"? Is the event simply when they compared their results to the observations? It would be good to know when, and where, the maximums occurred. In summary, at present the manuscript should not be accepted in to ACP. There are a number of critical shortcomings that may be addressed as discussed above. Those include, but are not limited to:

1. A much more straightforward and evenly presented model evaluation is needed. It should be conducted over the entire modeling domain for the entire modeling period, minus spin up days (which should be provided). Sub-regional evaluations can be presented, but they should be in addition to an overall evaluation. If subregional analysis are done for reduced periods, the specific criteria for the choice of period should be given. Saying they were "peak air pollution events" needs more definition (i.e., were they the highest ozone events in those cities that year?) It should not refer to excellent performance without appropriate citation. If it does not reach criteria or goal levels as laid out by either agency or peer-reviewed documents (preferably recent), they need to address this issue fully. 2. Precisely how the 39 cities were chosen should be specified. 3. In general, the text should be more precise over all. As noted above, "excellent" is one example. "peak photochemical period" is another. How the term "confidence in the accuracy of the model results" is used is another. 4. The discussion of how the model results align with empirical studies should be extended, and the differences not dismissed. Greater justification of the finding of natural gas being the major contributor is needed in light of past studies. If the goal really is to provide an analysis of UFP over the US, using a set of one 4-day period for each city without a traceable reason for the specific choice of period, does not accomplish the objective.

---

## Referee Comment (RC3) · Anonymous Referee #3 · 11 Oct 2018

In spite of their evident importance, there are very few data on ambient ultrafine particles, especially their mass concentrations and sources, in the scientific literature. The present papers aims to bring new insight into this topic using model simulations and concentrating on selected episodes in a large number of cities. In my opinion, this kind of a study is very welcome and should eventually be published. Before accepting the paper for publication, there are several issues that need to be clarified and revised in the paper. My detailed comments in this regard are given below.

Major issues

It is not entirely clear to me why the authors selected air pollution episodes lasting a

few days as the sole basis for estimating PM0.1 mass concentrations in different cities? Would a few days be much too short time period to get reliable information on different sources, and would selection of photochemical pollution episodes bias the importance of some sources over the others?

Since PM0.1 mass is the combined result of primary particle emissions (and nucleation) into this size range, and subsequent accumulatoin of secondary meterial by these particls, the authors should explain in more detail how they determined PM0.1 mass concentration (and the related source contribution) from their model simulations and what are the related uncertainties. There are several issues related to this. First, how many size bins the used model has in the sub-0.1 um size range and how close to the 0.1 is the border between the two nearest size bins? Now the authors only mention the number of size bin over the whole particle size range from 10 nm to 10 um (page 7). Second, what is the actual particle diameter used in model simulations? Mass measurements rely usually on aerodynamics diameters (impactors), while number size distribution measurements in the ultrafine size range rely usually on electric mobility diameters. These two diameters may differ substantially (up to a factor 2) for ambient aerosol particles, and the diameter used in a model can be either one of these two or something else. This is an important issue because PM mass size distributions often have a steep gradient at around 0.1 um, making the PM0.1 mass concentration very sensitive to the diameter chosen to represent the size 0.1 um. Third, the authors state that they do not care about nucleation because it only affects the particle number concentration but not the PM mass concentration. This is not true. Think, for example, a situation where 2 sources dominate the ultrafine particle number concentration: nucleation and a combustion source that produces particles with a peak diameter slightly below 100 nm. When these particle age in the atmosphere for a while and accumulate secondary material from the gas phase, those originating from nucleation tend to remain in the sub-100 nm size range while a big part of combustion particle may grow past 100 nm. As a result, whether or not to include nucleation also affects PM0.1 um mass. This issue should, at the very least, mentioned in the manuscript.

It is unclear to me how authors keep track on the different sources contributing to the PM0.1 mass concentration. I understand that keeping track particle numbers from different sources is possible, but how this is done for PM mass as a big fraction of it is formed secondarily in the atmosphere?

The authors use ozone and OC/BC concentrations in PM2.5 to evaluate their model. This is fine. However, it is clear that this sort of model evaluation does not guarantee that the model works well for PM0.1. While I do understand there are too few PM0.1 measurements around for a proper model evaluation in this respect, I still think that the authors should be more honest to state this explicitly in the manuscript (a good performance for ozone does, by no means, guarantee that also PM0.1 is simulated well).

I do not understand the last particle of the discussion before the conclusions, including equations 2 and 3 and figure 8. The obtained results misses a proper methodological description, and the actual results does not seem very helpful in the context of this paper. I would recommend removing this part of the analysis altogether from this paper.

Minor and technical issues

Section 2.3. Did the authors use real-meteorological data when calculating biogenic emissions using the MEGAN model? This is important because biogenic emissions are very sensitive to ambient temperatures. Please explain in the text.

What is the "Actual Max" for in the caption of figure 3? The given numbers are extremely accurate (3-5 digits) and well out of the scale of the relevant concentrations in these two sub-figures.

---

## Author Comment (AC1) · 2 Apr 2019

Predicted Ultrafine Particulate Matter Source Contribution across the Continental United States during Summer Time Air Pollution Events Venecek et al. 2018

ACP Anonymous Review – Author Response

Reviewer #1

1.) First, they choose 39 cities? What was the basis for the choice of cities? They appear to be most of the large cities, but it would still be good to know why those were chosen and others were not (fundamentally, did they choose the largest 39 cities, and

if not, why were others excluded? Having a research rational is important.).

Response: The cities selected for analysis are the largest urban regions across the United States that experienced 1-hr ozone concentrations above the level of 70 ppb in 2010. Many of these same cities have been analyzed in previous studies about urban air pollution throughout the continental US (Carter 1994, Carter 2007, Venecek et al 2018a and Venecek et al 2018b). These cities also form the basis for the ozone formation potential scales for VOCs developed by Carter (1994). These points have been clarified on Page 5, Line110-119 of the revised manuscript.

2.) The next very, very important issue is that they chose very limited time chunks in each of the cities, some overlapping, but a very odd collection. What motivated such a choice? If it was model-performance driven, than any statement about good model performance is not so relevant, or at least should be taken very carefully and explained, as the model performance metrics have not been designed for allowing the ability to choose specific locations and times when the model is performing adequately. (This would be akin to letting epidemiologists to go back and choose periods that specifically do or do not find associations. That would be viewed as bad practice and not acceptable.) It is my understanding, and I have checked this with colleagues, that the typical modeling with a state-of-the-science model like CAMx (which is also a regional chemical transport model, similar to UCD), it is applied over the domain over a chosen period, and then you look at all of the results for a model evaluation. (When applied in SIPS, both the location and time period are predetermined.) Thus, what is appears is done here, but not stated, is that it was applied over the continental US (or some similar domain) for some period(s), again not explicitly stated. The manuscript should state what was the actual modeling period used, or did they choose the specific simulation dates a priori for each city and just simulate those? This must be stated if it is the case. If that is indeed, the case, was the model started on the beginning date chosen, or was there a period allowed for the initial conditions to be minimized?

Response: The simulation dates were selected to capture the peak photochemical

air pollution episode at each location in the year 2010 identified by the measured peak ozone concentrations during that year. Each domain was simulated for one week with 3 days spin up and 4 days analysis such that the peak photochemical episode occurred on the last day of simulation. All simulation dates are stated in Table 1. Simulation dates were selected in regional clusters to focus on photochemical episodes driven by regional stagnation leading to the concentration of emissions from routine sources rather than extreme events driven by factors such as wildfires. The simulation dates therefore overlap for many cities within the same region. A figure has been added to the manuscript to illustrate how the dates overlap (figure 1 page 6).

We appreciate the reviewers concern that the episodes should be selected without regard to model performance criteria since this would indeed bias the findings. As described above, the simulation periods were selected using other independent criteria. These points have been clarified on page 5, lines 123-133 of the revised manuscript.

3.) Also, might the authors better justify choosing episodes beginning in March or October. The former seems a bit early, the latter a bit late. Were those the peak episodes that year?

Response: These were the peak ozone episodes that aligned in the south east/south United States. A recent ozone maximum incremental reactivity scale paper (Venecek et al 2018a) also utilized these dates and the average 1-hr max O3 can be found in the EPA AQS Data Mart.

4.) Are we to take a 3-4 day period of the year to represent the source impacts for the whole year? I would expect a rather different set of sources in the winter than the summer. This whole area is not explored or discussed.

Response: The simulated periods capture the maximum photochemical (peak) episodes across the entire US. The results therefore provide source apportionment of ultrafine particles during the peak photochemical period. The title of the manuscript has been changed to more clearly emphasize the focus of the paper (line 1).

Future studies will expand on the analysis to calculate source contributions for an entire year but this analysis is beyond the scope of the current paper.

5.) Why choose 2010 if UFP measurements are available other years? They might look at the recent Pandis paper (https://www.atmos-chem-phys.net/18/13639/2018/) doing a similar exercise over Europe. The choice of just peak ozone events may very well bias the assessment of source impacts on PM0.1 and PM2.5. Why not choose peak PM2.5 events?

Response: There are no consistent measurements of PM0.1 during any year at the majority of the locations simulated in the current study and so the choice of 2010 as a base year seems reasonable in order to leverage the large amount of background work that went into setting up the model episodes and verifying the model results in a related study (Venecek et al. 2018a).

We agree that simulating a full year with combustion for winter heating will lead to different source contributions for PM0.1. An expanded future study will consider a broader range of dates, but this analysis is beyond the scope of this initial study.

6.) The model performance part is also rather opaque and requires references. First the model performance should be brought in to the main manuscript, e.g., as done in most modeling papers, showing overall performance across the entire domain and modeling period (not just selected locations and periods). See Simon et al., (2014) for the metrics typically provided and a more complete discussion of model performance evaluation

Response: All model performance statistics have been brought into the main manuscript and a full comparison has been carried out between all predicted and measured gas and particle phase species (page 11, lines 233-242). Figure 3 and 4 have been added to show all FB and FE for all available monitors with lat/lon location available in the supporting information. Table 2 illustrates the percent of monitors within the entire modeling domain that meet US EPA criteria for 5 specific pollutants measured

throughout the EPA AQS datamart: CO, SO2, NO2, Ozone and PM2.5. As a quick summary of the new analysis, over 95% of the predictions compared to measurements across the entire US domain meet the EPA criteria.

7.) The working of the paragraph beginning on line 220 is also rather strange, it says "": : :MFB values lower than 0.15 and MFE values lower than 0.35 are considered the goal or "excellent" in model performance." Then they go on to say they do not meet them. First, I don't think I have seen EPA have "excellent" as a performance description associated with those levels. I was looking for a citation here (a citation to the specific EPA evaluation metrics is required, as well as adherence to the terminologies used. My understanding is that the current EPA guidance is found at https://www3.epa.gov/ttn/scram/guidance/guide/Draft_O3- PM-RH_Modeling_Guidance-2014.pdf, and looking through that document, I don't see them use the term excellent in terms of performance associated with any metric. (Indeed, a search of that document for the term "excellent" found only two occurrences, one in terms of protocol, another in terms of conceptual model.)

Response: The language on page 12, line 238 in the main manuscript describing model performance has been revised based on EPA guidance.

8.) That said, if the model does not meet the specific guidance levels, what does it meet? If the guidance was developed for regional scale modeling, without allowing selection by location and time, what does that imply here?). Second, if you do not meet them, what does that mean? Unlike the authors, I took the evaluation as not "building confidence in the accuracy of the model results: : :", but left me questioning it. I would very much recommend that the authors follow the EPA guidance (or other recent articles, e.g., those by the AQMEII initiative or Ramboll: Emery et al.) in terms of how to conduct, and report, model performance. Having now looked at the Ramboll study (JAWMA,doi/full/10.1080/10962247.2016.1265027), their ozone and PM performance do notmeet the "goal", nor fully meet the recommended levels for "criteria. One could also follow the approaches recommended by the AQMEII initiative.

[Figure]

Response: We apologize that the original version of the manuscript did not better emphasize the excellent model performance. Over 95% of the model predictions across the continental U.S. meet EPA criteria building confidence in the models predictions. The Reviewer's concerns about proper model evaluation are appreciated and we have strengthened the description of this aspect of the manuscript to give the readers a more complete view of model performance across all pollutants using all metrics recommended by Simon et al.

9.) Thinking more holistically, the proper approach here would be to apply the model to a whole year, or, if that is computationally infeasible (which should be stated), be applied to one month periods in each season, and the results from each of those months be given. If they only want to consider the peak photochemically-active periods, they should choose a three month period (or more, preferably) that will capture events in all the cities for that year.

Response: The reviewer is requesting a different study than the one that we performed. The focus of this current manuscript is the study of PM0.1 during the peak summer photochemical period across the United States in 2010. We have updated the title to reflect this focus "Predicted Ultrafine Particulate Matter Source Contribution across the Continental United States during Peak Summer Air Pollution Events". The following major conclusions of the paper will not change when an entire summer time period is simulated: (i) the majority of the PM0.1 is dominated by primary emissions; (ii) natural gas combustion is a major source of PM0.1 even though it makes minor contributions to PM2.5; (iii) there is significant variability in PM0.1 concentrations and source contributions between cities reflecting the different emissions in each city.

We agree that studies capturing seasonal averages and annual averages will be the next step now that this initial study on peak photochemical events has been completed. These studies will be the topics of future papers.

10.) I was a bit puzzled by the explanation given for Fig. 1. A 4 km grid is pretty fine,

and the mobile sources in a typical urban area like LA are pretty ubiquitous. There are a number of monitors in LA: do any of them not go to zero at night? If not, that might suggest a different issue The mismatch in the evening needs a bit more discussion and justification.

Response: The original manuscript did not incorporate the data from all 19 O3 monitors in the region around Los Angeles. The measurement data in Figure 1 (now Figure S1 in the revised manuscript) has been updated to reflect all available stations.

An error in the model wind fields was corrected in the revised version of the manuscript. This error had caused the winds in each row to advance by one column, effectively moving the winds over the Pacific Ocean over land for coastal California cities such as Los Angeles. The same error was corrected in all domains, but the effects were less severe at inland locations where winds were more uniform. All of the model results throughout the revised paper now reflect correct wind fields (all simulations were rerun).

The net result of the changes summarized above produce measured and predicted ozone concentrations that decrease to ∼zero during the evening hours. We thank the reviewer for pointing out the strange behavior in the original manuscript.

11.) They commit to providing the outdoor exposure fields. They should also provide the model and its inputs. I assume this is journal policy, but the authors should likewise commit

Response: The model itself and input data are available to collaborators through direct email request to the corresponding author. A statement to this effect has been added in the data availability section (page 27 line 466-468).

12.) A main conclusion of the paper is that natural gas is the main contributor to population-weighted exposure. This is a rather unique result and certainly requires more justification and discussion in light of the work that has shown via careful experiments that mobile sources and air craft are major sources of natural gas. Where is the

empirical evidence of natural gas being a main contributor and can they show that they have captured the contribution of those other two sources? Their explanation of why Lane et al., or Posner and Pandis is not sufficient to argue that the current results are reliable. How well does the current study capture the spatial dynamics found by the groups from USC (Sioutas), Harvard (Spengler) and UW (e.g., Atmospheric Environment Volume 139, August 2016, Pages 20-29) which tend to point the finger at mobile sources and aircraft, so much so, the latter claim that ultrafine particle counts can be used as a tracer for aircraft turbine emissions. Given the lack of empirical evidence, compounded with their not having done any evaluation of the ultrafine results against observations nationally, the speculative nature of this section suggests it should be removed, or couched in very different terms (i.e., noting the limitations, with a statement of the speculative nature). The question should be asked if there is sufficient evidence to support controls on a source based on the current analysis

Response: A recent study by Yu et al (2018) utilized the UCD-CIT CTM and compared predicted PM0.1 source contribution to PM0.1 CMB results using molecular markers (Xue et al 2018a) at multiple sites across California. The predictions from the UCD/CIT model were in good agreement with the CMB results for PM0.1 OC from gasoline (mobile), diesel (mobile), wood burning, meat cooking and "other sources". This comparison builds confidence in the accuracy of the regional UFP source predictions from the UCD/CIT model and the ability to properly represent contributions from mobile sources to regional PM0.1 concentrations.

The PM0.1 "other" category in the molecular marker calculation summarized by Xue et al. (2018) is composed of unresolved sources, but the UCD-CIT model at the core of the current manuscript can identify these sources. Major sources of the unresolved material identified by the UCD-CIT model include non-residential natural gas, aircraft and other source that were not tagged. The UCD-CIT model found that natural gas combustion is a significant source of PM0.1 OC in San Pablo, East Oakland, central Los Angeles, and Fresno where the predictions for contributions from mobile sources

were in good agreement with the CMB results.

Direct measurements of particle volatility in natural gas combustion exhaust made by Xue et al. (2018b) suggest that 70% of the natural gas combustion exhaust particles from residential sources (stoves and water heaters) evaporate when they are diluted in the atmosphere. Direct measurements indicated that particles emitted from engines operating on natural gas did not evaporate even at extremely high dilution ratios. The original version of the current manuscript specified that 70% of the particles from residential natural gas combustion sources would evaporate when diluted in the atmosphere, but it was assumed that particles emitted from commercial and industrial sources would not evaporate. Further review of typical commercial natural gas sources for space heating, water heating, etc suggested that these sources may be similar to residential natural gas combustion sources. Therefore, the model simulations in the revised manuscript were rerun while treating both residential and commercial natural gas combustion particles as semi-volatile (70% evaporation). The predicted contribution to PM0.1 from natural gas combustion particles decreases from 54% (original manuscript) to 33% (revised manuscript). Natural gas combustion particles are still important, but slightly less dominant in this revised treatment.

In summary, the current study uses all available empirical evidence to test and verify the predictions of natural gas combustion contributions to PM0.1 concentrations. The comprehensive comparisons to CMB studies in California show that the model calculations properly account for mobile source and food cooking contributions to PM0.1. The results across the rest of the US vary from location to location but are in general agreement with the relative important of mobile sources vs. other sources. We look forward to future datasets that perform PM0.1 CMB studies across the entire US, but do not believe that the findings from the current study should be delayed until those additional measurements have been completed.

13.) The performance evaluation example (e.g., in reference to the EPA goal/excellence criteria) is not the only place where a citation is needed. They state

that a number of other locations have PM0.1 levels above 2 ug/m3 (line 418). Then they go on to say there were sharper gradients in the observations. This would seem to contradict their findings/interpretation. More discussion needed

Response: The locations with PM0.1 greater than 2 $\mu$g m-3 were identified in the current study based on predictions from the UCD-CIT model. Likewise, the conclusion that sharper gradients were predicted PM0.1 vs. PM2.5 concentrations is based on UCD-CIT model predictions from the current study, not observations. These concluding statements summarize the findings of the model predictions, they do not seek to compare the model predictions to previous studies (previous sections of the paper are devoted to model performance evaluation). These statements do not contradict the findings or interpretation of the paper. We have clarified the sentence by adding the phrase "In the current study, predicted . . ." on line 447 of the revised manuscript.

14.) Again, they should compare their findings more directly to observations. In the summary, they state their analysis was for "peak photochemical periods during the year 2010." That is a rather strange way to characterize periods during which, in Atlanta, the 8-hr maximum ozone shown reached only about 50 ppb; I visually averaged, the actual value would be useful) in Cincinnati, about 60, in Los Angeles about 90, in New York about 85 ppb. I am not sure about the other locations, but I think the design value for LA in 2010 was about 120. In Cincinnati, it was about 0.079 (https://www3.epa.gov/airquality/greenbook/hbtcw.html), In Atlanta, about 80 ppb. The chosen periods would not appear to be "peak photochemical smog" periods

Response: The locations and dates correspond to periods when measured 1-hr ozone exceeded 70 ppb across the major geographical regions (south, south east, east, west etc.) in 2010. Figure 1 has been added to the manuscript to illustrate the ozone concentrations on the selected days. All monitor information (site lat/lon) can be found in the supporting information and obtained from the EPA AQS Datamart.

Note that the design value in 2010 is based on ozone measurements from 2008, 2009,

and 2010, with measured values from earlier years typically dominating the statistic during this time period. We believe that the episodes analyzed in the current study represent the peak air pollution events in the major US cities that are driven by routine emissions combined with stagnant meteorology. The revised manuscript thoroughly compares all available measurements of air pollution during the air pollution events.

15.) How different would Fig. 8 be if you simply used the local emissions? It is not apparent what this analysis adds and how it might be useful beyond simply using the inventoried sources

Response: The model application incorporates all major processes (emissions, transport, deposition, chemical reaction), which removes uncertainty of just using a local emission analysis. Given that this information is available, the authors are confused by the reviewers request to use an inferior analysis based only on the emissions inventory. Also note that Figure 8 has been moved to SI in response to another reviewer comment.

16.) Line 425 uses the word "consistent" after saying that the model could not resolve the observed mobile source peaks. While one can see what they may be trying to say, it should be said differently, and more precisely. One could easily have said, experiments have found peak ultrafine particle levels were tied to on-road mobile sources and aircraft emissions, though those three sources combined account for only 22%, and are thus inconsistent with the results here that identify natural gas combustion

Response: Changed text on page 27 line 454-458 to "On-road gasoline and diesel vehicles contributed on average 14% to regional PM0.1 even though peak contributions within 0.3 km of the roadway were not resolved by the 4 km grid cells. This is consistent with other studies that have found an exponential decrease in ultrafine particle concentrations outside of major roadways (Wang et al. 2011) due to the sharp gradient of PM0.1."

Minor:

1.) First line: should be "concentrations" Response: This has been updated in the text – page 2 line 12

2.) L 156: Missing ")" Response: Corrected.

3.) 4 km stated in both of the first lines of the Abstract.

Response: This has been updated in the text – page 2 line 15

4.) Fig. 2. Are these the individual, daily values for each site? If so, why are there not more circles? A bit more information in the figure caption would be useful.

Response: Speciated PM2.5 measurements are 24hr averages taken every 3 days or every 6 days depending on the city. Spin up days were not included in the comparison. All available comparison days were included in the analysis. The authors believe there are a sufficient number of data points (N>50) to properly evaluate the model performance.

5.) Figure 3. What do they mean by "air pollution event"? Is the event simply when they compared their results to the observations? It would be good to know when, and where, the maximums occurred.

Response: See figure 1 and response to previous comments describing criteria for selecting the regional maximum photochemical periods at each city in 2010.

  Reviewer #2

1. Page 2, lines 22-23. I would suggest removing "As expected." This minimizes the work, as if it were expected, why bother? Response: "As expected" has been removed on page 2 line 21-22 and page 13 line 319

2. In general, the paper could deal with some cleaning up of language, punctuation, etc. Examples Page 2, lines 31-32, use : : : for commercial use Page 3, line 57, should this be low birth weight Page 3, line 64, word national is not necessary, as monitors in the continental US are specified Page 4, lines 80 and 100 (and elsewhere) US or

U.S.? On page 22, line 410, United States is written out. In the SI, page 14, line 118, states is not capitalized. Page 4, line 85, add 'to' between exposure and ultrafine Page 7, line 156, missing a closing parenthesis Page 9, line 194 (and elsewhere including Fig 1 and Fig S2 caption), vs. not vs Page 17, lines 337-340 – Chang et al. (2004) measured: : : Add (2007) to Lane et al. In the SI, Figure S3 should appear after Table S2, as it is cited after Table S2 in the main text. SI, page 14, line 111, Figures compare, not compares

Response: The issues noted by the Reviewer above have been corrected and marked with yellow highlight in the main manuscript

3. In the abstract (Page 2, lines 35-37) and on page 20, lines 364-366), 'higher' and 'lower' ratios should be quantified. Is there a cutoff to determine higher versus lower based on the scatter plot shown in Figure 7?

Response: "Higher" ratio PM0.1/PM2.5 is anything higher than 0.10 and lower is anything lower than 0.05. Text has been added to the main manuscript on page 2 line 35

4. Page 3, line 60. I assume this should be surface area to volume ratio, not just surface area?

Response: This has been updated in the main manuscript page 3 line 61

5. Page 5, Table 1. Please provide more information about why these 39 cities were selected. Was it the availability of observations? Was it the number of O3 days above 70 ppb? As an example, why Charlotte and not Raleigh, NC? Or why Tulsa and not Oklahoma City, OK? Why were Pittsburgh and Chicago not included?

Response: The cities were selected based on the largest population centers across the US that experienced peak 1-hr ozone concentrations greater than 70 ppb in the year 2010. The locations generally correspond to previous studies that also looked at urban regions throughout the continental US (Carter 1994, Carter 2007, Venecek

2018a, Venecek 2018b). Some large population centers did not exceed the 70 ppb threshold and therefore were not included in the analysis. In general, the cities selected for analysis capture a cross section of urban populations across the US reflecting the diversity of emission sources.

6. Page 7, line 141. Please justify why nucleation is not considered. This is in line with a later comment about fraction of PM0.1 that is secondary versus primary.

Response: All of the simulations in the current study were rerun using nucleation based on the ternary nucleation (TN) mechanisms involving H2SO4-H2O-ammonia (NH3) (Napari et al, 2002). This mechanism has been applied in California with good agreement found between predicted and measured PM0.1 and N7 (Yu et al 2018). PM0.1 mass and source contributions in the current study did not change with the addition of nucleation, further confirming the conclusion that PM0.1 is driven by primary source contributions rather than nucleation.

8. Page 8, line 190. Even though it appears that secondary material is not a huge contribution to PM0.1, it would be appropriate to say 'also emit ultrafine particles and their precursors'

Response: Text has been added to the main manuscript page 11 line 226

11. Page 11, line 262. I recognize that the focus of this work is summer. However, would it be appropriate to highlight that the biomass contribution might be different in winter when wood burning for home heating could be a prevalent source of PM0.1 in colder regions?

Response: A sentence has been added at line 292-293 stating that wood combustion will make larger PM0.1 contributions during winter.

7. Section 2.3 and page 8, line 188. Please provide more information about monitors used and the comparison to model output. It says 'measurements averaged' – does this mean multiple monitors were used? Or was a single monitor compared to the model

output for the grid cell in which it resides? For cities with multiple monitors with a grid cell/domain, if multiple are used, it would be appropriate to include that information (perhaps in the SI).

Response: The model performance statistics for the re-generated simulations including nucleation have been updated. All monitors within a CBSA were compared to predictions. Figure 3 and 4 illustrate the MFB and MFE for all comparisons for all available gas and particle phase species (NO2, SO2, CO, O3 and PM2.5). All monitor information across the entire modeling domain has been added to the Supporting Information

9. Page 9. In reviewing Table S2, it appears that only one city does not meet the MFE for O3? If that is the case, it should be more specific on lines 199-200. It would be appropriate to provide the average O3 model performance statistics at this point. Then at the bottom of the page, the authors could discuss PM model performance statistics (and again, specify that only one city does not meet the MFE for PM model performance). Right now, it is slightly confusing to discuss O3, then PM, then both in terms of the averages.

Response: A more centralized presentation of all gas/particle phase model performance statistics have been added in section 3 (results). See response to Reviewer 1 comments 6 and 8.

10. Page 10, line 231. While I recognize that the submitted Yu et al. manuscript describes the 'good agreement' for PM0.1 modeling assessment in California, I think it could be summarized more quantitatively here in only one or two sentences.

Response: Yu et al PM0.1 source contribution for gasoline, diesel engines, food cooking, wood burning, and "other sources FE and FB were within EPA criteria of +/- 0.5 and 0.75, respectively added to page 12 line 259-263.

12. Page 13, Figure 2. How does the model convert from OM to OC? Does the two product model used (Carlton et al.) predict OM or OC? I thought it was OM, but if I

am mistaken please correct me. If a conversion is done to estimate OC based on the simulated OM, it would be appropriate to include this in the caption to Figure 2.

Response: The primary carbon variable tracked in model calculations is organic matter (OM), and the SOA model also predicts OM directly. These values must be converted to organic carbon (OC) for comparisons to measured values. Primary organic matter was converted to OC by dividing by a factor of 1.1. SOA components were converted to OC by dividing by a factor of 1.5. These points have been clarified on line 245 of the revised manuscript.

13. Page 14, Figure 3; Page 15, Figure 4. Would it be possible to somehow show on these figures the relative contribution of primary PM0.1 versus secondary PM0.1? This would truly drive home (and quantify) the relative contributions of direct emission versus in situ formation (I realize it is predominantly primary, but doing this would show it).

Response: Unfortunately, it is not possible to show the relative contributions on Figures 3 and 4. The primary vs. secondary fraction of PM2.5 and PM0.1 at each city location has been summarized in tables S7-S16 of the supporting information. A sentence summarizing this information has been added on line 313-315 of the revised manuscript.

14. Page 18, line 343. This paragraph does not seem necessary to me, as it focuses on previous work that simulated PNC, which as the authors point out in the nucleation discussion (see comment above), is not equivalent to PM0.1 (the focus of this work).

Response: Even though the focus of the current work is PM0.1, many researchers still use particle number concentration to describe UFPs. The discussion of how previous studies handled natural gas combustion emissions also explains why this source was not identified in previous studies. We therefore respectfully request that the paragraph be retained, but will defer to the Editor's judgement if the length of the paper is too long.

15. Page 22, Figure 8. A suggestion for improved readability: break up this figure into four panels by geographic region of the nation (since the focus is determining how cities in the same region compare – as discussed as 'regional clusters' on page 21).

Response: Note that the Figure has been moved to SI in response to comments by Reviewer 3. The authors feel that keeping the figure as one panel shows that PM0.1 source contribution across cities (even regional ones) do not correlate highly with one another and therefore emission control strategies should be tailored to each specific city. The text on page 66 of SI has been revised to describe the Figure.

16. SI, Page 13, line 95. The MFE given in the caption (0.67) is for O3? The MFE for PM given in the text is 0.75. Please specify both in the caption here. Also, note that the bold lines reflect cities that do not meet one of those criteria.

Response: As requested, updates have been made to the SI regarding all model performance statistics (Tables S1-S6 and Figure S1)

17. SI, Tables S3 and S4. These do not appear to be called out anywhere (if they were, and I missed it, I apologize). I assume this is the data that were used to create the vectors for the dot products? If so, that discussion is an appropriate place for them to be called out.

Response: The vector analysis has been moved to the SI and text has been added reflecting the use of these tables in the vector analysis. Page 65 line 20-21   Reviewer #3

1.) It is not entirely clear to me why the authors selected air pollution episodes lasting a few days as the sole basis for estimating PM0.1 mass concentrations in different cities? Would a few days be much too short time period to get reliable information on different sources, and would selection of photochemical pollution episodes bias the importance of some sources over the others?

Response: The simulation dates were selected to capture the peak photochemical

air pollution episode at each location in the year 2010 identified by the measured peak ozone concentrations during that year. Each domain was simulated for one week with 3 days spin up and 4 days analysis such that the peak photochemical episode occurred on the last day of simulation. All simulation dates are stated in Table 1. Simulation dates were selected in regional clusters to focus on photochemical episodes driven by regional stagnation leading to the concentration of emissions from routine sources rather than extreme events driven by factors such as wildfires. The simulation dates therefore overlap for many cities within the same region. A figure has been added to the manuscript to illustrate how the dates overlap (figure 1 page 6).

We agree that the current paper represents PM0.1 concentrations during a peak summer photochemical episode. Future studies will consider seasonal averages and annual averages, but this analysis is beyond the scope of the current manuscript.

2.) Since PM0.1 mass is the combined result of primary particle emissions (and nucleation) into this size range, and subsequent accumulatoin of secondary meterial by these particls, the authors should explain in more detail how they determined PM0.1 mass concentration (and the related source contribution) from their model simulations and what are the related uncertainties. There are several issues related to this. First, how many size bins the used model has in the sub-0.1 um size range and how close to the 0.1 is the border between the two nearest size bins? Now the authors only mention the number of size bin over the whole particle size range from 10 nm to 10 um (page 7). Second, what is the actual particle diameter used in model simulations? Mass measurements rely usually on aerodynamics diameters (impactors), while number size distribution measurements in the ultrafine size range rely usually on electric mobility diameters. These two diameters may differ substantially (up to a factor 2) for ambient aerosol particles, and the diameter used in a model can be either one of these two or something else. This is an important issue because PM mass size distributions often have a steep gradient at around 0.1 um, making the PM0.1 mass concentration very sensitive to the diameter chosen to represent the size 0.1 um. Third, the authors

state that they do not care about nucleation because it only affects the particle number concentration but not the PM mass concentration. This is not true. Think, for example, a situation where 2 sources dominate the ultrafine particle number concentration: nucleation and a combustion source that produces particles with a peak diameter slightly below 100 nm. When these particle age in the atmosphere for a while and accumulate secondary material from the gas phase, those originating from nucleation tend to remain in the sub-100 nm size range while a big part of combustion particle may grow past 100 nm. As a result, whether or not to include nucleation also affects PM0.1 um mass. This issue should, at the very least, mentioned in the manuscript.

Response: Nucleation using the ternary nucleation (TN) mechanisms involving H2SO4-H2O-ammonia (NH3) (Napari et al, 2002) has been added to the model configuration and the model simulations have been rerun. This mechanism has been applied in California with good agreement found between predicted and measured PM0.1 and N7 (Yu et al 2018). Nucleation did not significantly contribute to PM0.1 mass in the current study, and so the relative contributions from primary sources were unchanged due to the addition of nucleation.

Five (5) size bins equally spaced on a log diameter scale are used between 10 nm and 100 nm. The initial central diameters of each bin are: 12.6nm, 20nm, 32nm, 50nm, 79nm. Particle size bins "float" using the moving sectional approach. Condensation of secondary material causes particle growth while fresh emissions move the bin-averaged properties back towards the original emissions diameter. The model output therefore represents the competition between fresh emissions and atmospheric aging.

Number is tracked as an explicit variable for each moving size bin in the presence of all the major atmospheric processes (emissions, transport, deposition, gas-particle conversion, coagulation). The moving sectional approach naturally conserves particle number concentration since material is not transferred from one bin to another except through the relatively slow process of coagulation that mostly occurs between very

small particles and very large particles. The number concentration of the smaller bin involved in coagulation is reduced and the mass is transferred to the larger size bin. Number concentration is not the focus of the current manuscript, but additional details are provided by Yu et al. (2018).

3.) It is unclear to me how authors keep track on the different sources contributing to the PM0.1 mass concentration. I understand that keeping track particle numbers from different sources is possible, but how this is done for PM mass as a big fraction of it is formed secondarily in the atmosphere?

Response: Yu et al (2018), Ying et al., 2008b and Hu et al., 2017 provide a detailed description of how the model explicitly tracks mass in each particle size bin. A statement has been added to the main text of the current manuscript referencing those descriptions. In summary, the model explicitly tracks primary mass from different primary sources with an artificial tracer species. Tracer emissions are empirically set to be 1% of the total primary particle mass emitted from each source category. Tracers are carried through all major processes including transport, coagulation and deposition. The final tracer concentrations are directly proportional to the primary particle mass from the associated group.

Source contributions to PM0.1 SOA are tracked by tagging the emissions that feed into the chemical reaction mechanism. Reaction products inherit the tags from the parent compounds. Final semi-volatile reaction products that condense to the PM carry these same source tags allowing them to be quantified.

To be clear, 87% of the PM0.1 mass identified in the current study is primary, and so the tracer approach for primary emissions carries most of the source apportionment information.

4.) The authors use ozone and OC/BC concentrations in PM2.5 to evaluate their model. This is fine. However, it is clear that this sort of model evaluation does not guarantee that the model works well for PM0.1. While I do understand there are too few PM0.1

measurements around for a proper model evaluation in this respect, I still think that the authors should be more honest to state this explicitly in the manuscript (a good performance for ozone does, by no means, guarantee that also PM0.1 is simulated well).

Response: A recent study by Yu et al (2018) utilized the UCD-CIT CTM and compared predicted PM0.1 source contribution to CMB results using molecular markers (Xue et al 2018) in California. Source contributions to PM0.1 OC for gasoline (mobile), diesel (mobile), wood burning, meat cooking and "other" calculated using the UCD-CIT model and the molecular marker technique were in good agreement, which builds confidence in the accuracy of the UFP source predictions.

Good performance in California does not guarantee good performance across the entire US. The lack of data needed for model evaluation outside of California has been noted on Line 265 of the revised manuscript.

5.) I do not understand the last particle of the discussion before the conclusions, including equations 2 and 3 and figure 8. The obtained results misses a proper methodological description, and the actual results does not seem very helpful in the context of this paper. I would recommend removing this part of the analysis altogether from this paper.

Response: The current study identified that sources of PM2.5 and PM0.1 vary across major urban regions. The normalized dot product calculation allowed us to evaluate each city source contribution as a vector and quantitatively compare it to another city. The analysis found that few regional clusters were observed for PM0.1 source vectors, suggesting that emissions control programs may need to be tailored to each region.

The text describing Figure 8 has been clarified and the Figure has been moved to SI in the revised manuscript based on the Reviewer comment.

Minor and technical issues 1.) Section 2.3. Did the authors use real-meteorological

data when calculating biogenic emissions using the MEGAN model? This is important because biogenic emissions are very sensitive to ambient temperatures. Please explain in the text.

Response: MEGANv2.1 was configured with the same meteorology implemented into the UCD-CIT CTM. These met fields were determined using WRFv3.6. A statement has been added to the text to note this configurations (line 202)

2.) What is the "Actual Max" for in the caption of figure 3? The given numbers are extremely accurate (3-5 digits)

Response: Actual max is the predicted maximum PM2.5 and PM0.1 value ($\mu$g/m3).

---

## Referee Report (RR1)

This is the second time that I have reviewed this manuscript. My first step in this second review was to read the authors' response to all reviewer comments. The author responses were adequate and appropriate, in that they improved text that was unclear, answered important questions, and/or defended assumptions made. My second step was to re-read the manuscript to see if any additional questions arose. The points below are based on this re-reading of the manuscript.

Important Points/Questions/Clarifications
1. Are the concentrations presented only relevant for the surface level model cells?

2. On page 8, line 163, is N7 the number concentration of particles smaller than 7 micron? That is, does this subscript refer to size in micron as it does for subscripts on PM?

3. How sensitive are the OC and PM2.5 regressions to the OC:OM conversion factors used (page 11)? These values should include a citation. At this point, if the PM2.5 regression is included in the Supplemental Information, it should be mentioned here.

Editorial Points
In general, the information included in the introduction, methods, results and discussion are appropriate. That being said, I do believe that the manuscript could use some editing prior to publication. For example:

1. The first paragraph should be restructured – it could be three separate paragraphs: one on PM-health; one on PM0.1-health; one on surface area-health focused on smaller particles.

2. In some places O3 is used, in some places ozone is used. In several places, subscripts and superscripts are not used consistently. In some places units are expressed as ug/m3, in some places ug m-3 is used. On line 311, PM2/5 should be PM2.5.

3. In a couple of places (line 182; line 290), however is used as a coordinating conjunction. This requires either a semicolon or should be two separate sentences.

4. I do not believe that the most current version of the Supplemental Information of the manuscript was provided, as Tables/information called out in the main text do not appear, Figures/Tables are mis-numbered, Figure S1 is an exact copy of Figure 1 in the current version of the text… If the information that the main text says is in the new SI, it certainly sounds appropriate/adequate/important.

5. Why is there a * on LOUKY in Figure 2?

6. Correct spelling error in the x-axis title in Figure 6a. Measured, not mesaured.

---

## Author Response (AR2)

Re-review of Venecek et al., (2018) "Predicted Ultrafine… In the Continental United States"

The authors of the manuscript have now completed a lengthy reply to the first round of reviews, and while successfully addressing some of the issues, the responses and changes do not fully address others and introduce some added concerns, leading me to recommend against publication in ACP.

Comment #1: First, the manuscript, in my opinion, does not comport with ACP policy "If the data are not publicly accessible, a detailed explanation of why this is the case is required." Further, it states "Data do not comprise the only information which is important in the context of reproducibility. Therefore, Copernicus Publications encourages authors to also deposit software, algorithms, model code, video supplements, video abstracts, International Geo Sample Numbers, and other underlying material on suitable FAIR-aligned repositories/archives whenever possible. These materials should be referenced in the article and cited via a persistent identifier such as a DOI." They, instead, state "The model source code and input data are available to collaborators through direct email request to the corresponding author." Thus, it would appear, that the authors are unwilling to make the model and data available to others to assess if their results are reproducible, and as importantly, there are no errors in the data or the source code. On this latter point, it is important to note that it was via the initial review that an error that was identified: "An error in the model wind fields was corrected in the revised version of the manuscript. This error had caused the winds in each row to advance by one column, effectively moving the winds over the Pacific Ocean over land for coastal California cities such as Los Angeles. The same error was corrected in all domains, but the effects were less severe at inland locations where winds were more uniform. All of the model results throughout the revised paper now reflect correct wind fields (all simulations were rerun)." Without others being able to look over the inputs and the code, such errors can propagate forever. Indeed, one might expect that a similar error has been present in prior modeling using the same or predecessor codes to develop windfields. If this is the case, those journals must be contacted, and depending upon the journals discretion, erratum should be added to those prior publications. If research using fields where this is an issue have been used in health assessments and regulatory rule making (e.g., in California or by the US EPA), the agencies, themselves, should be contacted as to assess their further use. Given the potential importance of this issue, the authors need to reassure your journal and others, as well as the other users of this information, that a similar error was not present in the prior applications. Thus, in their response on-line, they would need to add an addendum to their current response stating that this issue was singular to this submission. While I realize that each journal has its own policies, I will note that the stated data policy by ACP is weaker than others like GMD, Science and Nature where the code would also have to be provided if requested to show that the results are reproducible and the code has no apparent errors.

*Response: All emissions inputs, spatial surrogates, field measurements used to generate model inputs and evaluate model results were obtained directly from the EPA database / model clearing house. These data are available to anyone wishing to re-create the inputs. The size and composition distribution profiles used to generate emissions of ultrafine particulate matter are publicly available in peer-reviewed journal articles published over the past several decades. Once again, these data are available to anyone wishing to re-create the inputs. If fellow researchers do not wish to go to the trouble of accessing the publicly available information to assemble the model inputs for themselves, then the authors are also*

*willing to collaborate as stated in the manuscript.  Please contact the corresponding author to discuss future collaborations.*

*All model systems have the potential for errors.  The process of continual model development combined with appropriate quality control checks identifies these errors so that they can be corrected.  In the current manuscript, the error in the wind fields was associated with an updated program that formatted WRF output fields for the UCDF/CIT air quality model.  This wind field error was not present in any previous manuscripts in ACP or in any other journal, since the prior version of the program did not contain the error.  We invite fellow researchers to inspect our source code and rerun model simulations as a quality control check if they wish to collaborate on ultrafine particle simulation studies.  Please contact the corresponding author to discuss future collaborations.*

*California and the US EPA do not base health assessments of regulatory rule making on a single model result.  A weight of evidence approach is always used in these efforts.  Independent measurements and/or model results would be needed to verify important findings before they are used in public policy decisions.*

Comment #2: A second rather important error, particularly since it was pointed out in the prior review, is the issue about model evaluation. There are no EPA guidelines, and the cited reference Boylan and Russell(1), does not represent EPA policy. I do not believe either of those individuals represents EPA. To assert that the cited manuscript represents EPA guidelines is quite wrong. The authors would be better guided to look at the Simon et al., paper (2) as most of the authors of that paper are from EPA, though they don't seem to suggest that they are establishing EPA guidelines. This was pointed out in the first review.

*Response: We acknowledge the point the Boylan and Russell manuscript does not represent official EPA policy even though these criteria are widely used for model evaluation studies.  We have modified the manuscript to remove references to "EPA" when discussing performance criteria and now reference the Emery et al (2017) paper as discussed in comments below.*

Comment #3: A further, extremely important issue is that the application of the Boylan et al. guidelines is also done incorrectly. That paper is only applicable to PM, not gaseous pollutants. To use the same suggested criteria for gaseous pollutants and PM is totally wrong. Further, they use this to suggest "excellent performance". This is where things get very concerning. They are misapplying suggested performance measures, to a subset of the model application (they keep saying 95% of the data: if you remove the worst 5% of the points, of course performance will improve: those metrics, I suspect, were not developed after removing the worst performing results: if so, the authors should so state), calling the guidelines "EPA", then saying "excellent" performance. In general, the authors tend to use very subjective superlatives rather casually and, I think, without justification.

*Response: We apologize for not correctly updating the model performance evaluation in Figure 3 during the last revision.  We have now updated Figure 3 and associated discussion based on the criteria suggested by Emery et al. (2017).  As discussed in the response to the following comment, over 95% of the predicted daily maximum ozone concentrations meet the performance criteria of <25% NME.*

*Please note especially that the worst points were not "removed" from the analysis. We provide quantitative comparisons between model performance and published guidelines so that the readers can reach their own conclusion about the accuracy of the simulations.*

Comment #4: A more realistic assessment of their model performance relative to others in North America is found from the Simon et al., article, which did not specify guidelines, or possibly a more recent article from researchers at Environ(2, 3) or the series from the AQMEII initiative, which has a more thorough set of approaches (4-11). The Simon/EPA article was a review of performance of ozone and PM modeling, again, not setting guidelines. Emery et al., reviewed the Simon et al., paper and others and provide the following suggested guidelines:

| | NMB | | NME | | r | |
|---|---|---|---|---|---|---|
| Species | Goal | Criteria | Goal | Criteria | Goal | Criteria |
| 1-hr or MDA8 Ozone | <±5% | <±15% | <15% | <25% | >0.75 | >0.50 |
| 24-hr PM2.5, SO4, NH4 | <±10% | <±30% | <35% | <50% | >0.70 | >0.40 |
| 24-hr NO3 | <±15% | <±65% | <65% | <115% | None | None |
| 24-hr OC | <±15% | <±50% | <45% | <65% | None | None |
| 24-hr EC | <±20% | <±40% | <50% | <75% | None | None |

*Response: The criteria listed in the first two lines of the table are the same criteria used in the updated manuscript.*

Comment #5: First, note that the bias for ozone suggested here as a goal is 5%, not the 60% they are using for fractional error. While it is difficult to do a direct comparison of fractional error to normalized mean error, it would appear that almost all of their results are performing worse than suggested. The NME suggested of 15% is, again, much tighter than the 75% they use for fractional error. Fundamentally, the ozone, performance, at least, does not appear to be very good vis a vis past studies. Considering these updated metrics, the performance looks to be substandard (though it is difficult to directly compare fractional metrics they use versus the normalized metrics used by Emery et al.). Just because it is a more recent paper, and that neither it nor the Boylan paper represent EPA policy, it is likely more appropriate to use the Emery article, and remove all references to being EPA guidelines. It is also very important that they specify precisely what data (e.g., hourly, no cut-off, specified days) is being used in the evaluation. I will note, the mean FB from Simon is only 0.03 for ozone, not the 0.6 they use. The typical results from this paper appear much higher. The Typical FE is only 0.22: not the 075 they show. A similar result is found when looking at other results from Simon: The mean results typically are much better than are used in the Venecek Table 2. If the authors do pursue publication, their results should be tabulated to provide statistics tht can be compared directly to the Simon et al. and Emery et al., papers, and should also consider the evaluation approaches and results from the AQMEII initiative(4-11).

*Response: We apologize for failing to correctly update the gas-phase model performance criteria in the last revision. Figure 3 and SI tables have been updated to reflect the criteria noted above (by Emery et al (2017)) for daily, 1-hr maximum ozone. Out of ~350 monitors, just over 95% met the performance criteria*

*mentioned above of <25% NME for max 1-hr O₃ values. In addition, all model predictions met the typical FE value of 0.22 also noted above. The authors believe the modeled O₃ values capture the peak photochemical episodes across the majority (95%) of monitor locations throughout the study domain. If the editor would like the author's to present more model performance statics we are happy to address these concerns in further revisions.*

Comment #6: There is also an issue in terms of what they show and how it is described. For example, the figure caption for Fig. 5 is "Figure 5 Predicted vs. Measured (a) Organic Carbon and (b) Elemental Carbon (μg m-3)" Are those the episode averages of EC and OC at each city? The daily values in each city? This is much, much different than showing the 24-hour values, for which performance is usually measured. In the manuscript and SI, they also need to be much more explicit as to what periods are being used for model performance.

*Response: Figure 5 shows the predicted 24-hr average EC or OC at each monitoring site location on each day that had available measurements. This is stated on the figure and included in the supporting information – however the Reviewer may have received an outdated SI (based on reviewer #1 comments).*

Comment #7: I note that the authors now stress that the results are only for a very short period during the summer. This makes their results of much less interest. How characteristic are UFP levels during a week in the summer vs. all year?

*Response: All versions of the paper have clearly stated that the focus was on a peak photochemical episode. PM concentrations are generally studied over short time periods of 24hrs and longer time periods such as an annual average. The current study is focused on short time periods during peak photochemical events.*

Comment #8: The OM to OC ratios used look low. A long-chain (say 15 C) alkane would have an OM to OC ratio of 1.17, so if you add even a little oxygen, it would be even higher. I seem to hear that secondary OM is more on the order of 2xOC. They should cite a source for the ratios used.

*Response: The author's agree with the reviewer than the OM to OC ratio should be slightly higher and have recalculated OC to have a ratio of 1.2 (Russell 2003). Figure 5 on page 18 has been updated in the main manuscript and citation added to the references.*

Comment #8: In summary, I would have to recommend against publication in its current form as detailed above. The lack of conformity with journal guidelines on data (and further, I would argue, not providing the data and underlying code would look bad for the journal unless a very good reason, i.e., a specific restriction on the data/code, in which case another code should likely be used for scientific studies), the misuse and characterization of model performance metric comparison, and using a really short period without providing a great reason for why such short periods are of scientific interest, are all important. In particular, I don't think ACP should be publishing a paper with major mistakes pointed out and that does not appear to comport to their guidelines on data and data and code availability of other leading journals.

*Response: We respect the Reviewers opinion, but respectfully disagree with their conclusion.*

References

1.      Boylan JW & Russell AG (2006) PM and Light Extinction Model Performance Metrics, Goals, and Criteria for Three-Dimensional Air Quality Models. Atmos. Environ. 40(26):4946-4959.

2.      Simon H, Baker KR, & Phillips S (2012) Compilation and interpretation of photochemical model performance statistics published between 2006 and 2012. Atmos. Environ. 61:124-139.

3.      Emery C, et al. (2017) Recommendations on statistics and benchmarks to assess photochemical model performance. J. Air Waste Manage. Assoc. 67(5):582-598.

4.      Galmarini S, Rao ST, & Steyn DG (2012) AQMEII: An International Initiative for the Evaluation of Regional-Scale Air Quality Models - Phase 1 Preface. Atmos. Environ. 53:1-3.

5.      Hogrefe C, et al. (2015) Annual application and evaluation of the online coupled WRF-CMAQ system over North America under AQMEII phase 2. Atmos. Environ. 115:683-694.

6.      Koo B, et al. (2015) Chemical transport model consistency in simulating regulatory outcomes and the relationship to model performance. Atmos. Environ. 116:159-171.

7.      Nopmongcol U, et al. (2012) Modeling Europe with CAMx for the Air Quality Model Evaluation International Initiative (AQMEII). Atmos. Environ. 53:177-185.

8.      Rao ST, Galmarini S, & Puckett K (2011) Air Quality Model Evaluation International Initiative (AQMEII) Advancing the State of the Science in Regional Photochemical Modeling and Its Applications. Bulletin of the American Meteorological Society 92(1):23-30.

9.      Rao ST, et al. (2014) Air Quality Model Evaluation International Initiative (AQMEII): A Two-Continent Effort for the Evaluation of Regional Air Quality Models. Air Pollution Modeling and Its Application Xxii, NATO Science for Peace and Security Series C-Environmental Security, eds Steyn DG, Builtjes PJH, & Timmermans RMA), pp 455-462.

10.     Solazzo E, Galmarini S, Bianconi R, & Rao ST (2014) Model Evaluation for Surface Concentration of Particulate Matter in Europe and North America in the Context of AQMEII. Air Pollution Modeling and Its Application Xxii, NATO Science for Peace and Security Series C-Environmental Security, eds Steyn DG, Builtjes PJH, & Timmermans RMA), pp 375-379.

11.     Wang K, et al. (2015) A multi-model assessment for the 2006 and 2010 simulations under the Air Quality Model Evaluation International Initiative (AQMEII) Phase 2 over North America: Part II. Evaluation of column variable predictions using satellite data. Atmos. Environ. 115:587-603.